# Targeted Advertising on Social Networks Using Online Variational Tensor Regression (revised ver.2.1)

## Abstract

This paper is concerned with online targeted advertising on social networks. The main technical task we address is to estimate the activation probability for user pairs, which quantifies the influence one user may have on another towards purchasing decisions. This is a challenging task because one marketing episode typically involves a multitude of marketing campaigns/strategies of different products for highly diverse customers. In this paper, we propose what we believe is the first tensor-based contextual bandit framework for online targeted advertising. The proposed framework is designed to accommodate any number of feature vectors in the form of multi-mode tensor, thereby enabling to capture the heterogeneity that may exist over user preferences, products, and campaign strategies in a unified manner. To handle inter-dependency of tensor modes, we introduce an online variational algorithm with a mean-field approximation. We empirically confirm that the proposed TensorUCB algorithm achieves a significant improvement in influence maximization tasks over the benchmarks, which is attributable to its capability of capturing the user-product heterogeneity.

## 1 Introduction

Online targeted advertising is one of the most interesting applications of machine learning in the Internet age. In a typical scenario, a marketing agency chooses a set of "seed" users from the nodes (i.e. users) of a social graph, and makes certain offers (e.g. coupons, giveaways, etc.), with the expectation that the seed users will influence their followers and spread the awareness on the product(s) or service(s) being promoted. An important question of interest is how to maximize the total purchases accrued over multiple marketing campaign rounds under a fixed budget (i.e. the number of seed users per round). This task is commonly referred to as (online) influence maximization (IM) in the machine learning community.

A key quantity of interest here is the ***activation probability*** $\{p_{i,j}\}$, where $p_{i,j}$ is the probability of user $i$ influencing user $j$ into buying the products being advertised. Since $\{p_{i,j}\}$ is unknown a priori, we are to repeatedly update the estimate after each marketing round, starting from a rough initial estimate based, for example, on demographic information. Many trials and errors are unavoidable especially in the beginning. After a sufficient number of trials, however, we can expect to have systematically better estimates for $\{p_{i,j}\}$. These characteristics make the *contextual bandits* (CB) framework (Abe et al., 2003; Li et al., 2010; Bouneffouf et al., 2020) a relevant and attractive solution approach. Here, the "bandit arms" correspond to the seed users to be selected. The "context" corresponds to information specific to the products being advertised and the users being targeted. The "reward" would be the number of purchases attained as a result of the influence of the selected seed users.

There are two mutually interacting sub-tasks in IM as discussed in the literature: One is how to choose the seed users when given $\{p_{i,j}\}$; The other is how to estimate $\{p_{i,j}\}$ given a seed selection algorithm, which is the focus of this paper. The approaches to the latter task can be further categorized into *direct* and *latent* modeling approaches. The direct approaches mainly leverage graph connectivity combined with simple features such as the number of purchases. Since transaction history is typically very sparse, the latent modeling approach has been attracting increasing attention recently. In this category, two major approaches

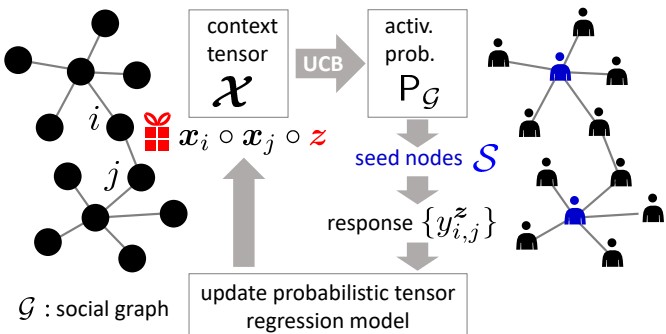

Figure 1: Overview of `TensorUCB` (simplest case). The context tensor encodes the features of user pairs and product/marketing strategy, and is used to estimate the activation probability matrix $\mathsf{P}_{\mathcal{G}} = [p_{i,j}^{\boldsymbol{z}}]$. Based on observed user responses, the estimation model is updated.

have been proposed to date. One is *regression-based* (Vaswani et al., 2017; Wen et al., 2017) and the other is *factorization-based* (Wu et al., 2019; Wang & Chen, 2017; Barbieri et al., 2013). Although encouraging results have been reported in these works, there is one important limitation that restricts their usefulness in practice: *Absence of capability to incorporate product features.* This is critical in practice since marketing campaigns typically include many different products and strategies applied to a diverse population, and different types of products are expected to follow different information diffusion dynamics.

To address this issue, we propose `TensorUCB`, a general tensor-based contextual bandit framework. Unlike the prior works, we use a tensor to represent the context information (*"context tensor"*), which makes it possible to handle *any* number of feature vectors in principle. Figure 1 illustrates our problem in the simplest setting. The user context tensor $\boldsymbol{\mathcal{X}}$ is formed from three feature vectors in this simplest case: user feature vectors of the $i$- and $j$-th users and a product feature vector $\boldsymbol{z}$. By construction, the model accommodates *edge-level* feedback that can depend on the product type. Then, the activation probability matrix $\mathsf{P}_{\mathcal{G}} \triangleq [p_{i,j}^{\boldsymbol{z}}]$ is estimated as a function of $\boldsymbol{\mathcal{X}}$. Here we used $p_{i,j}^{\boldsymbol{z}}$ instead of $p_{i,j}$ to show the dependence on $\boldsymbol{z}$. We formalize this task as online probabilistic tensor regression. As shown later, it can effectively capture the heterogeneity over product types using a low-rank tensor expansion technique. We integrate probabilistic estimation from tensor regression with the upper confidence bound (UCB) policy in a way analogous to the LinUCB algorithm (Li et al., 2010).

To the best of our knowledge, this is the first proposal of a contextual bandit framework extended to context tensors and applied to the task of IM. Our empirical results show that the proposed methods outperform baseline algorithms in the presence of product heterogeneity.

## 2  Related work

Prior works relevant to this paper can be categorized into three major areas: CB-based IM approaches, tensor bandits, and tensor regression.

**CB-based IM**  Following the pioneering works by Valko et al. (2014) and Chen et al. (2016) that framed IM as an instance of the bandit problem, a few approaches have been proposed to incorporate contextual information. Vaswani et al. (2017) proposed `DILinUCB`, a contextual version of IM bandits, which uses user contextual features to learn $p_{i,j}$ with linear regression. Wu et al. (2019) proposed `IMFB`, which exploits matrix factorization instead of linear regression. Unlike our work, in which a single susceptibility tensor $\boldsymbol{\mathcal{W}}$ is shared by all the nodes, their approaches give latent parameters to each network node, and thus, tends to require more exploration. Wen et al. (2017) proposed another regression-based approach `IMLinUCB` using edge-specific features, which can be difficult to obtain in practice. There also exist prior works that attempt to capture product features in addition to the user features. Sarıtaç et al. (2016) proposed `COIN`, which assumes a predefined partition of product features and does not directly use the users' context vectors. Our

framework automatically learns multiple patterns in different products as well as different users through multi-rank tensor regression. Chen et al. (2015) consider a topic distribution for the seed selection task, not for learning $\{p_{i,j}^{\boldsymbol{z}}\}$.

**Tensor bandits** Apart from IM, bandits with structured arms are an emerging topic in the bandit research community. The majority of the studies consider the bilinear setting, which can be solved through low-rank matrix estimation or bilinear regression (Kveton et al., 2017; Zoghi et al., 2017; Katariya et al., 2017; Lu et al., 2018; Hamidi et al., 2019; Jun et al., 2019b; Lu et al., 2021). However, it is not clear how they can be extended to general settings having more than two contextual vectors which we are interested in. Azar et al. (2013) is among the earliest works that used higher order tensors in bandit research. However, their task is transfer learning among a multitude of bandits, which is different from ours. Recently, Hao et al. (2020) proposed a tensor bandit framework based on the Tucker decomposition (Kolda & Bader, 2009). However, their setting is *not contextual* and is not applicable to our task. Specifically, in our notation, their reward model is defined solely for $\boldsymbol{\mathcal{X}} = \boldsymbol{e}_{j_1}^1 \circ \cdots \circ \boldsymbol{e}_{j_D}^D$, where $\boldsymbol{e}_{j_l}^l$ is the $d_l$-dimensional unit basis vector whose $j_l$-th element is 1 and otherwise 0. As a result of this binary input, the coefficient tensor $\boldsymbol{\mathcal{W}}$ is directly observable through the response $u$. In other words, the task is *not* a supervised learning problem anymore in contrast to our setting. To the best of our knowledge, our work is the first proposal of variational tensor bandits in the contextual setting.

**Tensor regression** We also believe this is the first work of contextual tensor bandits allowing an arbitrary number of tensor modes. For generic tensor regression methods, limited work has been done on *online* inference of *probabilistic* tensor regression. Most of the existing probabilistic tensor regression methods (e.g. (Zhao et al., 2014; Imaizumi & Hayashi, 2016; Guhaniyogi et al., 2017; Ahmed et al., 2020)) require either Monte Carlo sampling or evaluation of complicated interaction terms, making it difficult to directly apply them to online marketing scenarios. In particular, they do not provide an analytic form of predictive distribution, which is desirable to make the UCB framework applicable. We provide a tractable online updating equation based on a variational mean-field approximation. To the best of our knowledge, `TensorUCB` is among the first works that explicitly derived an online version of probabilistic tensor regression.

## 3 Problem Setting

In the online influence maximization (IM) problem on social networks, there are three major design points, as illustrated in Fig. 1:

- Estimation model for $y_{i,j}^{\boldsymbol{z}}$ (user $j$'s response (purchase etc.) by user $i$'s influence for a product $\boldsymbol{z}$).

- Scoring model for $p_{i,j}^{\boldsymbol{z}}$ (the probability that user $i$ activates user $j$ for a product $\boldsymbol{z}$).

- Seed selection model to choose the $K$ most influential users, given $\{p_{i,j}^{\boldsymbol{z}}\}$ and a social graph $\mathcal{G}$.

This paper deals with the first and the second tasks alone, following the existing IM literature (Wu et al., 2019; Vaswani et al., 2017; Wen et al., 2017; Sarıtaç et al., 2016). We formalize the first task as online probabilistic regression that takes a tensor as the contextual input (Sec. 4). The second task is handled by integrating the derived probabilistic model with the idea of UCB (Sec. 5). The third task is not within the scope of this paper. It takes care of the combinatorial nature of the problem and is known to be NP-hard (Kempe et al., 2003). We assume to have a black-box subroutine (denoted by `ORACLE`) that produces a near-optimal solution for a given $\{p_{i,j}\}, K$, and a social graph $\mathcal{G}$. In our experiments we use an $\eta$-approximation algorithm (Golovin & Krause, 2011) proposed by Tang et al. (2014).

Although the use of $\{p_{i,j}^{\boldsymbol{z}}\}$ implies the independent cascade (IC) model (Kempe et al., 2003) as the underlying diffusion process, we do not explicitly model the dynamics of information diffusion. Instead, we learn the *latent* quantity $p_{i,j}^{\boldsymbol{z}}$ as a proxy for diffusion dynamics among the users. This is in contrast to the *direct* approaches (Bhagat et al., 2012; Li et al., 2013; Morone & Makse, 2015; Lei et al., 2015; Lu et al., 2015), as mentioned in Introduction.

Table 1: Main mathematical symbols.

| symbol | definition |
|---|---|
| $y_{i,j}^{\boldsymbol{z}}$ | Binary user response for the event $i \Rightarrow j$ for a product $\boldsymbol{z}$. |
| $\bar{u}_{i,j}^{\boldsymbol{z}}$ | Expected score (real-valued) for $y_{i,j}^{\boldsymbol{z}}$. |
| $p_{i,j}^{\boldsymbol{z}}$ | Activation probability of the event $i \Rightarrow j$ for a product $\boldsymbol{z}$. |
| $\boldsymbol{\mathcal{X}}$ | Context tensor $\boldsymbol{\mathcal{X}} = \boldsymbol{\phi}_1 \circ \boldsymbol{\phi}_2 \circ \cdots \circ \boldsymbol{\phi}_D$ (Eq. (4)) with $\phi_l$ in $\mathbb{R}^{d_l}$ for $l = 1, \dots, D$. |
| $\phi_1$ | Source user's feature vector, which is $\boldsymbol{x}_i$ in the event $i \Rightarrow j$. |
| $\phi_2$ | Target user's feature vector, which is $\boldsymbol{x}_j$ in the event $i \Rightarrow j$. |
| $\phi_3$ | Product feature vector (typically denoted by $\boldsymbol{z}$). |
| $\boldsymbol{w}^{l,r}$ | Coefficient vector for $\phi_l$ of the $r$-th tensor rank. |
| $\bar{\boldsymbol{w}}^{l,r}$ | Posterior mean of $\boldsymbol{w}^{l,r}$ (Eq. (15)). |
| $\boldsymbol{\Sigma}^{l,r}$ | Posterior covariance matrix of $\boldsymbol{w}^{l,r}$ (Eq. (14)). |

## 3.1 Data model

In addition to a social graph $\mathcal{G} = (\mathcal{V}, \mathcal{E})$, where $\mathcal{V}$ is the set of user nodes ($|\mathcal{V}|$ is its size) and $\mathcal{E}$ is the set of edges ($|\mathcal{E}|$ is its size), we consider two types of observable data. The *first* is the contextual feature vectors. There are two major types of feature vectors. One is the user feature vector $\{\boldsymbol{x}_i \mid i \in \mathcal{V}\}$ while the other is the product feature vector denoted by $\boldsymbol{z}$. Let us denote by $i \Rightarrow j$ the event that "user $i$ activates user $j$ (into buying a product $\boldsymbol{z}$)." The contextual information of this event is represented by a tuple $(\boldsymbol{x}_i, \boldsymbol{x}_j, \boldsymbol{z})$. There can be other feature vectors representing campaign strategies, etc. In general, we assume that an activation event is characterized by $D$ contextual feature vectors $\boldsymbol{\phi}_1 \in \mathbb{R}^{d_1}, \dots, \boldsymbol{\phi}_D \in \mathbb{R}^{d_D}$, where $d_1$, etc., denote the dimensionality. All the feature vectors are assumed to be real-valued column vectors.

In Fig. 1, we illustrated the case where $\boldsymbol{\phi}_1 = \boldsymbol{x}_i$, $\boldsymbol{\phi}_2 = \boldsymbol{x}_j$, and $\boldsymbol{\phi}_3 = \boldsymbol{z}$ for the user pair $(i, j)$ and a product having the feature vector $\boldsymbol{z}$. As summarized in Table 1, we will always allocate $\boldsymbol{\phi}_1, \boldsymbol{\phi}_2, \boldsymbol{\phi}_3$ to the source user, the target user, and the product feature vectors, respectively. Creating the feature vectors is not a trivial task in general. See Section 7.1 for one reasonable method.

The *second* observable is the users' response, denoted by $y_{i,j}^{\boldsymbol{z}} \in \{0, 1\}$ for the event $i \Rightarrow j$ for a product $\boldsymbol{z}$. $y_{i,j}^{\boldsymbol{z}} = 1$ if $i \Rightarrow j$ has occurred, and $y_{i,j}^{\boldsymbol{z}} = 0$ otherwise. Although activation is not directly measurable in general, a widely-used heuristic is a time-window-based method (Barbieri et al., 2013). Specifically, we set $y_{i,j}^{\boldsymbol{z}} = 1$ if user $j$ bought the product after actively communicating with user $i$ within a certain time window. Active communications include "likes," retweeting, and commenting, depending on the social networking platform. The size of the time window is determined by domain experts and is assumed to be given.

## 3.2 Activation probability estimation problem

We consider the situation where a fixed number (denoted by $K$) of seed users are chosen in each campaign round ("budgeted IM"). The seed nodes may have a different number of connected nodes, as illustrated in Fig. 2. Thus, the dataset from the $t$-th marketing round takes the following form:

$$\{(\boldsymbol{\phi}_{t(k),1}, \dots, \boldsymbol{\phi}_{t(k),D}, y_{t(k)}) \mid k = 1, \dots, n_t\}, \tag{1}$$

where $n_t = \sum_{i \in \mathcal{S}_t} n_i^{\text{out}}$, $\mathcal{S}_t$ is the set of seed users chosen in the $t$-th marketing round ($|\mathcal{S}_t| = K$) and $n_i^{\text{out}}$ is the number of outgoing edges of the $i$-th node. In this expression, the identity of node pairs and the product is implicitly encoded by $k$. In our solution strategy, the estimation model is updated as soon as a new sample comes in. Hence, it is more useful to "flatten" $(t, k)$ into a single "time" index $\tau$ when considering all the samples obtained up to the current time $\tau$, denoted as

$$\mathcal{D}_{1:\tau} \triangleq \{(\boldsymbol{\phi}_{\tau',1}, \dots, \boldsymbol{\phi}_{\tau',D}, y_{\tau'}) \mid \tau' = 1, \dots, \tau\}. \tag{2}$$

As a general rule, we use a subscript ($t(k)$ or $\tau$) to denote an *instance* of a random variable.

Our main task is to estimate the activation probability matrix $\mathsf{P}_{\mathcal{G}} \triangleq [p_{i,j}^{\boldsymbol{z}}]$ as a function of the contextual feature vectors $\boldsymbol{\phi}_1, \dots, \boldsymbol{\phi}_D$, where $\boldsymbol{\phi}_1 = \boldsymbol{x}_i, \boldsymbol{\phi}_2 = \boldsymbol{x}_j$ and $\boldsymbol{\phi}_2 = \boldsymbol{z}$ are assumed and we define $p_{i,j}^{\boldsymbol{z}} = 0$ for

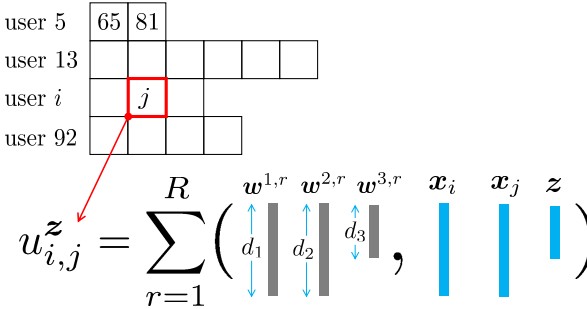

Figure 2: Illustration of data structure in one marketing round and the prediction model, corresponding to Eq. (3) (the noise term is omitted for simplicity). $\{5, 13, i, 92\}$ is the set of seed nodes here. $K = 4$ and $D = 3$ are assumed.

disconnected node pairs. As mentioned at the beginning of this section, the task is divided into two steps. The *first* step (estimation model) is to learn a regression function to predict $y_{i,j}^{\boldsymbol{z}}$ from $\boldsymbol{\phi}_1, \ldots, \boldsymbol{\phi}_D$. We call the output of the regression function the *response score*, and denote it by $u_{i,j}^{\boldsymbol{z}} \in \mathbb{R}$ to distinguish it from the binary response:

$$u_{i,j}^{\boldsymbol{z}} = H_{\boldsymbol{\mathcal{W}}}(\boldsymbol{\phi}_1, \boldsymbol{\phi}_2, \ldots, \boldsymbol{\phi}_D) + (\text{noise}), \tag{3}$$

where we have used the convention $\boldsymbol{\phi}_1 = \boldsymbol{x}_i, \boldsymbol{\phi}_2 = \boldsymbol{x}_j$, and $\boldsymbol{\phi}_3 = \boldsymbol{z}$, as mentioned before. $H_{\boldsymbol{\mathcal{W}}}$ is a parametric model with $\boldsymbol{\mathcal{W}}$ being a random variable called the susceptibility tensor (defined in the next section). To systematically treat the uncertainty in the user response, we wish to learn a *probability distribution* of $\boldsymbol{\mathcal{W}}$ and $u_{i,j}$ explicitly, and eventually derive their *updating rule* to get a renewed estimate in every marketing round $t$. As described in the next section, the noise term is assumed to be Gaussian.

In the *second* step (scoring model), once the distribution of $u_{i,j}^{\boldsymbol{z}}$ is obtained, the activation probability $p_{i,j}^{\boldsymbol{z}} \in [0, 1]$ is computed not only with the expectation $\bar{u}_{i,j}^{\boldsymbol{z}}$ but also with the variance through an appropriate mapping function that reflects the UCB policy.

## 4 Online variational tensor regression

When activation $i \Rightarrow j$ occurs, we naturally assume that the activation probability depends on the feature vectors of the user pair *and* the product. One straightforward approach in this situation is to create a concatenated vector and apply, e.g., the LinUCB algorithm (Li et al., 2010), in which $H_{\boldsymbol{\mathcal{W}}}$ is the linear regression function. However, it is well-known that such an approach is quite limited in its empirical performance. For concreteness, consider the $D = 3$ case in Fig. 1 again. The main issue is that it amounts to treating $\boldsymbol{x}_i, \boldsymbol{x}_j, \boldsymbol{z}$ separately hence failing to model their *interactions*: $H_{\boldsymbol{\mathcal{W}}}$ in this approach would be $\boldsymbol{w}_1^\top \boldsymbol{x}_i + \boldsymbol{w}_2^\top \boldsymbol{x}_j + \boldsymbol{w}_3^\top \boldsymbol{z}$, and fitting the regression coefficients $\boldsymbol{w}_1, \boldsymbol{w}_2, \boldsymbol{w}_3$ would result in giving the most weight on generally popular user and product types. This is not useful information in online advertising, as we are interested in analyzing what kind of *affinity* there might be in a specific combination of user pairs and products. The proposed tensor-based formulation allows dealing with such interactions while keeping the computational cost reasonable with a low-rank tensor approximation.

### 4.1 Tensor regression model

We instead assume to have the *context tensor* in the form

$$\boldsymbol{\mathcal{X}} = \boldsymbol{\phi}_1 \circ \boldsymbol{\phi}_2 \circ \cdots \circ \boldsymbol{\phi}_D, \tag{4}$$

where $\circ$ denotes the direct product. For example, in the case shown in Fig. 1, the $(i_1, i_2, i_3)$-th element of $\boldsymbol{\mathcal{X}}$ is given by the product of three scalars: $[\boldsymbol{x}_i \circ \boldsymbol{x}_j \circ \boldsymbol{z}]_{i_1,i_2,i_3} = x_{i,i_1} x_{j,i_2} z_{i_3}$, where the square bracket denotes the operator to specify an element of tensors. As mentioned before, $\boldsymbol{\phi}_1, \boldsymbol{\phi}_2$ and $\boldsymbol{\phi}_3$ correspond to

the source user, target user, and product feature vectors, respectively. The other feature vectors $\phi_4, \ldots, \phi_D$ can represent marketing campaign strategies, etc. Note that Eq. (4) includes the regression model in *bilinear* bandits (e.g. (Jun et al., 2019a)) and *non*-contextual tensor bandits (Hao et al., 2020) as special cases. Specifically, bilinear models can handle only the $D = 2$ case while our model can handle $D \geq 3$. *Non*-contextual tensor regression cannot accommodate the feature vectors $\{\phi_l \in \mathbb{R}^{d_l} \mid l = 1, \ldots, D\}$.

For the regression function $H_{\mathcal{W}}$ in Eq. (3), we employ a tensor regression model as

$$H_{\mathcal{W}}(\phi_1, \ldots, \phi_D) = (\mathcal{W}, \mathcal{X}), \qquad (\mathcal{W}, \mathcal{X}) \triangleq \sum_{i_1, \ldots, i_D} \mathcal{W}_{i_1, \ldots, i_D} \mathcal{X}_{i_1, \ldots, i_D}, \tag{5}$$

where $(\cdot, \cdot)$ denotes the tensor inner product, and we call the regression coefficient $\mathcal{W}$ the *susceptibility tensor*. For tractable inference, we employ the CP (canonical polyadic) expansion (Cichocki et al., 2016; Kolda & Bader, 2009) of order $R$, which simplifies Eq. (5) significantly:

$$\mathcal{W} = \sum_{r=1}^{R} \boldsymbol{w}^{1,r} \circ \boldsymbol{w}^{2,r} \circ \cdots \circ \boldsymbol{w}^{D,r}, \qquad u_{i,j} = \sum_{r=1}^{R} \prod_{l=1}^{D} \phi_l^{\top} \boldsymbol{w}^{l,r} + (\text{noise}), \tag{6}$$

where $^{\top}$ denotes the transpose. In the second equation above, the r.h.s. now involves only the vector inner products. There are two important observations to note here. *First*, this particularly simple form is due to the specific approach of the CP expansion we employ. Other tensor factorization methods such as Tucker and tensor-train do not yield simple expressions like Eq. (6), making the UCB analysis intractable. *Second*, with $R > 1$, it has multiple regression coefficients for each tensor mode $l$. This flexibility provides the potential to capture the characteristics of multiple product types, unlike vector-based linear regression.

Figure 2 illustrates one marketing round with $K = 4$ and $D = 3$. Each seed user has a few connected users. For example, the 5th user is a "friend" of the 65th and 81st users. For a user pair $(i, j)$, the response score $u_{i,j}$ is computed from $\phi_1 = \boldsymbol{x}_i$, $\phi_2 = \boldsymbol{x}_j$, and $\phi_3 = \boldsymbol{z}$ through Eq. (6).

### 4.2  Variational learning of susceptibility tensor

One critical requirement in CB-based IM is the ability to handle stochastic fluctuations of the user response. Here we provide a fully probabilistic online tensor regression model.

As the first step, let us formalize a batch learning algorithm, assuming that all the samples up to the $\tau$-th "time" are available at hand under the flattened indexing as in Eq. (2). Define $\mathcal{X}_\tau \triangleq \phi_{\tau,1} \circ \cdots \phi_{\tau,D}$. As mentioned before, $\phi_{\tau,1}$ and $\phi_{\tau,2}$ are used for the node feature vectors, serving as the proxy for the node indexes. We employ Gaussian observation and prior models, which follow the standard CB approach except tensor-based parameterization:

$$p(u \mid \mathcal{X}, \mathcal{W}, \sigma) = \mathcal{N}(u \mid (\mathcal{W}, \mathcal{X}), \sigma^2), \qquad p(\mathcal{W}) = \prod_{l=1}^{D} \prod_{r=1}^{R} \mathcal{N}(\boldsymbol{w}^{l,r} \mid \boldsymbol{0}, \mathsf{I}_{d_l}), \tag{7}$$

where $p(\cdot)$ symbolically represents a probability distribution and $\mathcal{N}(\cdot \mid (\mathcal{W}, \mathcal{X}), \sigma^2)$ denotes Gaussian with mean $(\mathcal{W}, \mathcal{X})$ and variance $\sigma^2$. Also, $u \in \mathbb{R}$ is the user response score (at any time and user pair), and $\mathsf{I}_d$ is the $d$-dimensional identity matrix. Since $\sigma^2$ is assumed to be given and fixed, which is a common assumption in the bandit literature, $\mathcal{W}$ is the only model parameter to be learned.

Despite the apparent simplicity of Eq. (6), inter-dependency among the parameter vectors $\{\boldsymbol{w}^{l,r}\}$ makes exact inference intractable. To address this issue, we introduce *variational tensor bandits* featuring variational Bayes (VB) inference (Bishop, 2006). The key assumption of VB is to assume the posterior distribution in a factorized form. In our case, the posterior of the susceptibility tensor $\mathcal{W}$ is assumed to be:

$$Q(\mathcal{W}) = Q(\{\boldsymbol{w}^{l,r}\}) = \prod_{l=1}^{D} \prod_{r=1}^{R} q^{l,r}(\boldsymbol{w}^{l,r}). \tag{8}$$

We determine the distribution $\{q^{l,r}\}$ by minimizing the Kullback-Leibler (KL) divergence:

$$Q = \arg\min_Q \text{KL}[Q\|Q_0], \qquad \text{KL}[Q\|Q_0] \triangleq \int \prod_{l=1}^{D}\prod_{r=1}^{R} d\boldsymbol{w}^{l,r}\, Q(\boldsymbol{\mathcal{W}}) \ln \frac{Q(\boldsymbol{\mathcal{W}})}{Q_0(\boldsymbol{\mathcal{W}})}, \tag{9}$$

where $Q_0(\boldsymbol{\mathcal{W}})$ is the true posterior. We, of course, do not know the exact form of $Q_0(\boldsymbol{\mathcal{W}})$, but we do know that it is proportional to the product between the observation and prior models by Bayes' theorem:

$$Q_0(\boldsymbol{\mathcal{W}}) \propto p(\boldsymbol{\mathcal{W}}) \prod_\tau p(y_\tau \mid \boldsymbol{\mathcal{X}}_\tau, \boldsymbol{\mathcal{W}}, \sigma). \tag{10}$$

Equation (9) is a functional optimization problem. Fortunately, the Gaussian assumption allows us to find an analytic form for the posterior. The result is simple: $q^{l,r}(\boldsymbol{w}^{l,r}) = \mathcal{N}(\boldsymbol{w}^{l,r} \mid \bar{\boldsymbol{w}}^{l,r}, \boldsymbol{\Sigma}^{l,r})$, where

$$\bar{\boldsymbol{w}}^{l,r} = \sigma^{-2}\boldsymbol{\Sigma}^{l,r}\sum_\tau \boldsymbol{\phi}_{\tau l}\beta_\tau^{l,r}y_\tau^{l,r}, \qquad \boldsymbol{\Sigma}^{l,r} = [\,\sigma^{-2}\sum_\tau \boldsymbol{\phi}_{\tau,l}\boldsymbol{\phi}_{\tau,l}^\top \gamma_{\tau,l} + \mathsf{I}_{d_l}]^{-1}. \tag{11}$$

Here we have defined

$$\beta_\tau^{l,r} \triangleq \prod_{l'\neq l}\boldsymbol{\phi}_{\tau,l'}^\top \bar{\boldsymbol{w}}^{l',r}, \quad y_\tau^{l,r} \triangleq y_\tau - \sum_{r'\neq r}(\boldsymbol{\phi}_{\tau,l}^\top \bar{\boldsymbol{w}}^{l,r'})\beta_\tau^{l,r'}, \quad \gamma_{\tau,l} \triangleq \prod_{l'\neq l}\boldsymbol{\phi}_{\tau,l'}^\top \langle \boldsymbol{w}^{l',r}(\boldsymbol{w}^{l',r})^\top\rangle_{\backslash(l,r)}\boldsymbol{\phi}_{\tau,l'}, \tag{12}$$

where $\langle\cdot\rangle_{\backslash(l,r)}$ is the partial posterior expectation excluding $q^{l,r}$. Derivation of Eqs. (11)-(12) is straightforward but needs some work. See Appendix A for the details.

## 4.3 Mean-field approximation and online updates

Equations (11)-(12) have mutual dependency among the $\{\boldsymbol{w}^{l,r}\}$ and need to be performed iteratively until convergence. This is numerically challenging to perform in their original form. In Eq. (11), $\gamma_{\tau,l} \in \mathbb{R}$ plays the role of the sample weight over $\tau$'s. Evaluating this weight is challenging due to the matrix inversion needed for $\boldsymbol{\Sigma}^{l',r}$. For faster and more stable computation suitable for sequential updating scenarios, we propose a mean-field approximation $\langle \boldsymbol{w}^{l',r}(\boldsymbol{w}^{l',r})^\top\rangle_{\backslash(l,r)} \approx \bar{\boldsymbol{w}}^{l',r}(\bar{\boldsymbol{w}}^{l',r})^\top$, which gives $\gamma_{\tau,l} = (\beta_\tau^{l,r})^2$. Intuitively, the mean-field approximation amounts to the idea "think of the others as given (as their mean) and focus only on yourself." Using this, we have a simple formula for $\boldsymbol{\Sigma}^{l,r}$:

$$\boldsymbol{\Sigma}^{l,r} = \left[\,\sigma^{-2}\sum_\tau \left(\beta_\tau^{l,r}\boldsymbol{\phi}_{\tau,l}\right)\left(\beta_\tau^{l,r}\boldsymbol{\phi}_{\tau,l}\right)^\top + \mathsf{I}_{d_l}\right]^{-1}. \tag{13}$$

Unlike the crude approximation that sets the other $\{\boldsymbol{w}^{l,r}\}$ to a given constant, $\boldsymbol{w}^{l,r}$'s are computed iteratively over all $l, r$ in turn, and are expected to converge to a mutually consistent value. The variance is used for comparing different edges in the UCB framework. The approximation is justifiable since the mutual consistency matters more in our task than estimating the exact value of the variance. In Sec. 7, we will confirm that the variational tensor bandits significantly outperforms the baseline even under these approximations.

Now let us derive the online updating equations. Fortunately, this can be easily done because $\bar{\boldsymbol{w}}^{l,r}$ in Eq. (11) and $\boldsymbol{\Sigma}^{l,r}$ in Eq. (13) depend on the data only through the summation over $\tau$. For any quantity defined as $A_{\tau+1} \triangleq \sum_{s=1}^{\tau} a_s$, we straightforwardly have an update equation $A_{\tau+1} = A_\tau + a_\tau$ in general. Hence when a new sample $(\boldsymbol{\mathcal{X}}_\tau, y_\tau)$ comes in: First, $\boldsymbol{\Sigma}^{l,r}$ can be updated as

$$(\boldsymbol{\Sigma}^{l,r})^{-1} \leftarrow (\boldsymbol{\Sigma}^{l,r})^{-1} + (\beta^{l,r}/\sigma)^2\boldsymbol{\phi}_{\tau,l}\boldsymbol{\phi}_{\tau,l}^\top, \qquad \boldsymbol{\Sigma}^{l,r} \leftarrow \boldsymbol{\Sigma}^{l,r} - \frac{\boldsymbol{\Sigma}^{l,r}\boldsymbol{\phi}_{\tau,l}\boldsymbol{\phi}_{\tau,l}^\top\boldsymbol{\Sigma}^{l,r}}{(\sigma/\beta^{l,r})^2 + \boldsymbol{\phi}_{\tau,l}^\top\boldsymbol{\Sigma}^{l,r}\boldsymbol{\phi}_{\tau,l}}, \tag{14}$$

where the second equation follows from the Woodbury matrix identity (Bishop, 2006). Second, for the posterior mean $\bar{\boldsymbol{w}}^{l,r}$, with the updated $\boldsymbol{\Sigma}^{l,r}$, we have

$$\boldsymbol{b}^{l,r} \leftarrow \boldsymbol{b}^{l,r} + \boldsymbol{\phi}_{\tau,l}\beta^{l,r}y_\tau^{l,r}, \qquad \bar{\boldsymbol{w}}^{l,r} = \sigma^{-2}\boldsymbol{\Sigma}^{l,r}\boldsymbol{b}^{l,r}. \tag{15}$$

Equations (14)-(15) are computed over all $(l,r)$ until convergence. Note that when $R = D = 1$, these update equations essentially derive the ones used in `LinUCB` (Li et al., 2010) as a special case.

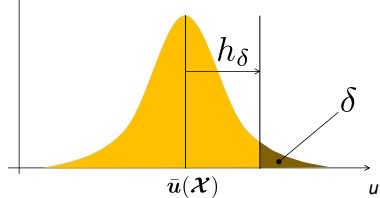

Figure 3: Illustration of Gaussian tail probability and its upper confidence bound.

## 5 Tensor UCB Algorithm

This section presents `TensorUCB`, based on the online probabilistic tensor regression framework in Sec. 4.

### 5.1 Predictive distribution

With the posterior distribution $Q(\mathcal{W}) = \prod_{l,r} q^{l,r}$, we can obtain the predictive distribution of the user response score $u$ for an *arbitrary* context tensor $\mathcal{X} = \phi_1 \circ \cdots \circ \phi_D$ as

$$p(u \mid \mathcal{X}, \mathcal{D}_{1:\tau}) = \int \mathcal{N}(u \mid (\mathcal{W}, \mathcal{X}), \sigma^2) \, Q(\mathcal{W}) \, \mathrm{d}\mathcal{W}. \tag{16}$$

Despite $Q(\mathcal{W})$ being a factorized Gaussian, this integration is intractable. We again use the mean-field approximation when applying the Gaussian marginalization formula (see, e.g., Sec. 2.3.3 of (Bishop, 2006)). The resulting predictive distribution is also a Gaussian distribution $p(u|\mathcal{X}, \mathcal{D}_t) = \mathcal{N}(y \mid \bar{u}(\mathcal{X}), \bar{s}^2(\mathcal{X}))$ with

$$\bar{u}(\mathcal{X}) = (\bar{\mathcal{W}}, \mathcal{X}) = \sum_{r=1}^{R} \prod_{l=1}^{D} (\bar{w}^{l,r})^\top \phi_l, \quad \bar{s}^2(\mathcal{X}) = \sigma^2 + \sum_{r=1}^{R} \sum_{l=1}^{D} (\beta^{l,r} \phi_l)^\top \Sigma^{l,r} (\beta^{l,r} \phi_l). \tag{17}$$

Notice that the predictive mean $\bar{u}(\mathcal{X})$ is simply the inner product between the posterior mean $\bar{\mathcal{W}}$ and the input tensor $\mathcal{X}$, which is a typical consequence of the mean-field approximation. We also see that the predictive variance $\bar{s}^2$ depends on the context tensor $\mathcal{X}$, and some users/products may have greater uncertainty in the expected score $\bar{u}$. We leave the detail of the derivation of Eq. (17) to Appendix B.

### 5.2 Upper confidence bound

To transform $\bar{u}$ into the activation probability, we adopt the well-known UCB strategy. Thanks to the predictive distribution being Gaussian, we can straightforwardly provide the upper confidence bound corresponding to a tail probability.

We start with Markov's inequality that holds for any non-negative random variable $v$ and any $r > 0$:

$$\mathbb{P}(v \geq r) \leq \frac{\langle v \rangle}{r}, \tag{18}$$

where $\mathbb{P}(\cdot)$ is the probability that the argument holds true and $\langle \cdot \rangle$ is the expectation. Although the response score $u$ can be negative, we can use Markov's inequality by setting $v = \mathrm{e}^{\lambda u}$:

$$\mathbb{P}(\mathrm{e}^{\lambda u} \geq \mathrm{e}^{\lambda h}) = \mathbb{P}(u \geq h) \leq \frac{\langle \mathrm{e}^{\lambda u} \rangle}{\mathrm{e}^{\lambda h}}. \tag{19}$$

Here we note that $\langle \mathrm{e}^{\lambda u} \rangle$ is the definition of the moment generating function, and a well-known analytic expression is available for the Gaussian distribution (17):

$$\langle \mathrm{e}^{\lambda u} \rangle = \exp\left( \lambda \bar{u}(\mathcal{X}) + \frac{1}{2} \lambda^2 \bar{s}^2(\mathcal{X}) \right). \tag{20}$$

---

**Algorithm 1** `TensorUCB` for contextual influence maximization ($D = 3$ case)

---
**Input:** $K, \sigma, R$ and $c > 0$. Subroutine `ORACLE`.
Initialize $\{\boldsymbol{\Sigma}^{l,r}, \boldsymbol{w}^{l,r}\}$ and $\{p_{i,j}\}$.
**for** $t = 1, 2, ..., T$ **do**
    Receive product context $\boldsymbol{\phi}_3 = \boldsymbol{z}$ of this round.
    $\mathcal{S}_t \leftarrow \texttt{ORACLE}(\{p_{i,j}^{\boldsymbol{z}}\}, K, \mathcal{G})$
    Receive response from users connected to $\forall i \in \mathcal{S}_t$.
    **for** $k = 1, \ldots, n_t$ **do**
        Retrieve user feature vectors $\boldsymbol{\phi}_1, \boldsymbol{\phi}_2$ from $k$.
        Update $\{\bar{\boldsymbol{w}}^{l,r}\}$ and $\{\boldsymbol{\Sigma}^{l,r}\}$ with $(\boldsymbol{\mathcal{X}}_{t(k)}, y_{t(k)})$.
    **end for**
    **for** $(i, j) \in \mathcal{E}$ **do**
        Set $\boldsymbol{\phi}_1 = \boldsymbol{x}_i, \; \boldsymbol{\phi}_2 = \boldsymbol{x}_j$, and $\boldsymbol{\mathcal{X}} = \boldsymbol{\phi}_1 \circ \boldsymbol{\phi}_2 \circ \boldsymbol{\phi}_3$
        Compute $p_{i,j}^{\boldsymbol{z}} = \text{proj}\left(\bar{u}(\boldsymbol{\mathcal{X}}) + \texttt{UCB}(\boldsymbol{\mathcal{X}})\right)$
    **end for**
**end for**

---

The inequality (19) now reads

$$\mathbb{P}(u \geq h) \leq \exp\left(\lambda(\bar{u}(\boldsymbol{\mathcal{X}}) - h) + \frac{1}{2}\lambda^2 \bar{s}^2(\boldsymbol{\mathcal{X}})\right). \tag{21}$$

Here, recall that $\lambda$ is *any* positive number. The idea of Chernoff bound is to exploit the arbitrariness of $\lambda$ to get the tightest bound. It is an elementary calculus problem to get the minimum of the r.h.s. of Eq. (21). The minimum is achieved at $\lambda = (h - \bar{u})/\bar{s}^2$, yielding the Chernoff bound of Gaussian:

$$\mathbb{P}(u \geq h) \leq \exp\left(-\frac{(h - \bar{u}(\boldsymbol{\mathcal{X}}))^2}{2\bar{s}^2(\boldsymbol{\mathcal{X}})}\right). \tag{22}$$

Now let us assume that the tail probability $\mathbb{P}(u \geq h)$ on the l.h.s. equals to $\delta$, and denote the corresponding upper bound by $h_\delta + \bar{u}$, as illustrated in Fig. 3. Solving the equation $\delta = \exp\left(-h_\delta^2/[2\bar{s}^2(\boldsymbol{\mathcal{X}})]\right)$, we have

$$h_\delta = \sqrt{-2 \ln \delta} \times \bar{s}(\boldsymbol{\mathcal{X}}). \tag{23}$$

This is the upper confidence bound we wanted. Since $\sigma^2$ has been assumed to be a constant and $(\beta^{l,r}\boldsymbol{\phi}_l)^\top \boldsymbol{\Sigma}^{l,r}(\beta^{l,r}\boldsymbol{\phi}_l) \geq 0$ in Eq. (17), it suffices to use

$$p_{i,j}^{\boldsymbol{z}} = \text{proj}\left(\bar{u}(\boldsymbol{\mathcal{X}}) + \texttt{UCB}(\boldsymbol{\mathcal{X}})\right), \qquad \texttt{UCB}(\boldsymbol{\mathcal{X}}) \triangleq c \sum_{r=1}^{R} \sum_{l=1}^{D} \sqrt{(\beta^{l,r}\boldsymbol{\phi}_l)^\top \boldsymbol{\Sigma}^{l,r}(\beta^{l,r}\boldsymbol{\phi}_l)}, \tag{24}$$

where we remind the reader that $\boldsymbol{\phi}_1 = \boldsymbol{x}_i, \; \boldsymbol{\phi}_2 = \boldsymbol{x}_j$, and $\boldsymbol{\phi}_3 = \boldsymbol{z}$ in $\boldsymbol{\mathcal{X}}$. Also, $\text{proj}(\cdot)$ is a (typically sigmoid) function that maps a real value onto $[0, 1]$, and $c > 0$ is a hyperparameter. Again, $D = R = 1$ reproduces LinUCB (Li et al., 2010).

### 5.3 Algorithm summary

We summarize the proposed `TensorUCB` algorithm in Algorithm 1 in the simplest setting with $D = 3$. `ORACLE` has been defined in Sec. 3. Compared with the existing budgeted IM works using linear contextual bandits, `TensorUCB` has one extra parameter, $R$, the rank of CP expansion (6). $R$ can be fixed to a sufficiently large value within the computational resource constraints, typically between 10 and 100. As shown in Fig. 6 later, the average regret tends to gradually improve as $R$ increases up to a certain value. Except for the first several values as in Fig. 6, changes are typically not drastic.

As for the other three "standard" parameters: The budget $K$ is determined by business requirements. $\sigma^2$ is typically fixed to a value of $\mathcal{O}(1)$ such as 0.1. Note that $\sigma^2$ is also present in other linear contextual bandit

frameworks but they often ignore it by assuming unit variance. $c$ needs to be chosen from multiple candidate values (see Sec. 7), which is unavoidable in UCB-type algorithms. For initialization of the online algorithm, $\mathbf{\Sigma}^{l,r}$ is typically set to $\mathsf{I}_{d_l}$ and $\boldsymbol{w}^{l,r}$ can be the vector of ones. $\{p_{i,j}^{\boldsymbol{z}}\}$ can be non-negative random numbers for connected edges and 0 otherwise.

Since our algorithm does not need explicit matrix inversion, the complexity per update can be evaluated as $\mathcal{O}(RDd^2)$, where $d \triangleq \max_l d_l$. Note that, if we vectorized $\boldsymbol{\mathcal{W}}$ to use standard vector-based inference algorithms, the complexity would be at least $\mathcal{O}((\prod_l d_l)^2)$, which can be prohibitive. The vectorized model also hurts interpretability as it breaks natural groupings of the features.

## 6   Regret analysis

We leverage the predictive distribution (17) to evaluate the regret bound of `TensorUCB`. Let $f(\mathcal{S}, \mathsf{P}_{\mathcal{G}})$ be the total number of target users activated by a selected seed set $\mathcal{S}$ based on the activation probabilities $\mathsf{P}_{\mathcal{G}}$. Suppose that `ORACLE` (see Sec. 3 for the definition) is an $\eta$-approximation algorithm (Golovin & Krause, 2011). In the CB-based IM literature, a scaled version of regret is typically used as the starting point (Wen et al., 2017; Vaswani et al., 2017; Chen et al., 2016):

$$R_T^{\eta} = \frac{1}{\eta} \sum_{t=1}^{T} \mathbb{E}[f(\mathcal{S}^*, \mathsf{P}_{\mathcal{G}}^*) - f(\mathcal{S}_t, \mathsf{P}_{\mathcal{G}}^*)], \tag{25}$$

where $\mathsf{P}_{\mathcal{G}}^* = [p_{i,j}^{*\boldsymbol{z}}]$ is the ground truth of the activation probability matrix and $\mathcal{S}^* = \texttt{ORACLE}(\mathsf{P}_{\mathcal{G}}^*, K, \mathcal{G})$. One campaign round is assumed to handle only one type of product. The expectation $\mathbb{E}(\cdot)$ is taken over the randomness of $\boldsymbol{z}$ and the other non-user contextual vectors (i.e., $\boldsymbol{\phi}_l$ with $l \geq 3$) as well as seed selection by the `ORACLE`. Let $\mathsf{P}_{t,\mathcal{G}} = [p_{t,i,j}^{\boldsymbol{z}}]$ be the estimated activation probability matrix at the $t$-th round.

To derive a regret bound in our setting, we need to make a few assumptions. The first assumption (A1) is called the bounded smoothness condition, which is commonly used in the bandit IM literature. We assume that there exists a constant $B$ such that

$$f(\mathcal{S}_t, \mathsf{P}_{t,\mathcal{G}}) - f(\mathcal{S}_t, \mathsf{P}_{\mathcal{G}}^*) \leq B \sum_{i \in \mathcal{S}_t} \sum_{j \sim i} |p_{t,i,j}^{\boldsymbol{z}} - p_{i,j}^{*\boldsymbol{z}}|, \tag{26}$$

for any $\mathsf{P}_{\mathcal{G}}^*, \mathcal{S}_t$, where the second summation for $j$ runs over the nodes connected to the selected seed node $i$. The second assumption (A2) is another commonly used condition called the monotonicity condition. For a seed user set $\mathcal{S}$ and a product $\boldsymbol{z}$, the monotonicity states that, if $p_{i,j}^{\boldsymbol{z}} \leq p_{i,j}^{\prime \boldsymbol{z}}$ for all $i \in \mathcal{S}$ and their connected nodes $j$, we have $f(\mathcal{S}, \mathsf{P}_{\mathcal{G}}) \leq f(\mathcal{S}, \mathsf{P}_{\mathcal{G}}')$. The third assumption (A3) is $\|\beta_{t(k)}^{l,r} \boldsymbol{\phi}_{t(k),l}\| \leq 1$, $\forall l, r, t, k$, which can be always satisfied by rescaling the feature vectors. Finally, the fourth assumption (A4) is about variability due to the variational Bayes and mean-field approximations. Unlike vector-based linear regression, no exact analytic solution is known in *probabilistic* tensor regression, and its rigorous theoretical analysis in the context of *contextual* bandits is still an open problem. In what follows, we ignore their variability, assuming that $\texttt{UCB}(\boldsymbol{\mathcal{X}})$ in Eq. (24) is exact.

Now we state our main result on the regret bound:

**Theorem 1.** *Under the assumptions (A1)-(A4) stated above and a condition*

$$c \geq DR\sqrt{\frac{Kd\ln\left(1 + \frac{TK}{d\sigma^2}\right)}{\ln\left(1 + \frac{1}{\sigma^2}\right)}} + \max_{l,r}\|\boldsymbol{w}^{l,r}\|_2, \tag{27}$$

*the upper regret bound of* `TensorUCB` *is given by*

$$R_T^{\eta} \leq \mathcal{O}\left(\frac{cB}{\eta}|\mathcal{V}|DR\sqrt{\frac{TKd\ln\left(1 + \frac{TK}{d\sigma^2}\right)}{\ln\left(1 + \frac{1}{\sigma^2}\right)}}\right) \tag{28}$$

*with probability at least $1 - \delta$.*

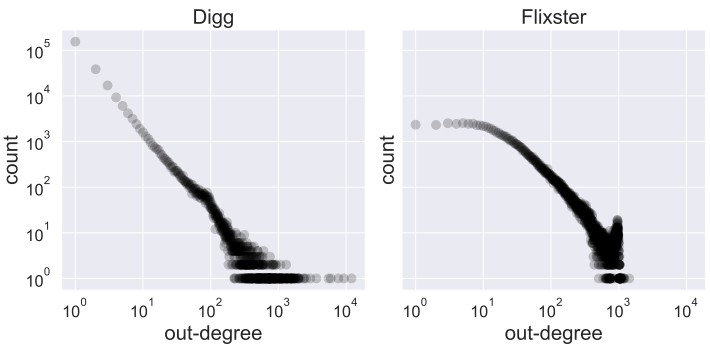

Table 2: Network statistics, where $n^{\mathrm{out}}(m)$ denotes the $m$-th percentile of the out-degree.

|  | Digg | Flixster |
|---|---|---|
| $|\mathcal{V}|$ | 2 483 | 29 384 |
| $|\mathcal{E}|$ | 75 895 | 371 722 |
| $n^{\mathrm{out}}(50)$ | 1 | 9 |
| $n^{\mathrm{out}}(90)$ | 8 | 154 |
| $n^{\mathrm{out}}(99)$ | 101 | 2064 |

Figure 4: Distribution of out-degrees.

The proof is given in Appendix C. Although the complexity of this bound can depend on the assumed scenario, it is at least comparable to the ones reported in the literature: `IMFB` (Wu et al., 2019), `DILinUCB` (Vaswani et al., 2017), and `IMLinUCB` (Wen et al., 2017) reported the regret bounds of $\mathcal{O}(d|\mathcal{V}|^{\frac{5}{2}}\sqrt{T})$, $\mathcal{O}(d|\mathcal{V}|^2\sqrt{T})$, and $\mathcal{O}(d|\mathcal{V}|^3\sqrt{T})$, respectively, which hold under a similar condition to Eq. (27).

## 7 Experiments

This section reports on empirical evaluation of `TensorUCB`. Our goal is to illustrate how it captures users' heterogeneity as the tensor rank $R$ increases (Fig. 6), to show its advantage in the presence of product/user heterogeneity (Fig. 5), and to examine its computational overhead (Fig. 7).

### 7.1 Datasets

We used two publicly available datasets with significantly different levels of heterogeneity in products and users: *Digg* (Hogg & Lerman, 2012a;b) records users' voting history to posted news stories and has $|\mathcal{V}| = 2\,843$, $|\mathcal{E}| = 75\,895$, and 1 000 stories. *Flixster* (Zafarani & Liu, 2009) records users' movie rating history and has $|\mathcal{V}| = 29\,384$, $|\mathcal{E}| = 371\,722$, and 100 movies. Notice that the number of products is as large as 1 000. These are real-world datasets where naive product-wise modeling is unrealistic. Figure 4 compares their distributions of out-degrees (i.e., the number of outgoing edges). In both datasets, the distribution is highly skewed, as also seen in the out-degree percentiles summarized in Table 2. The out-degree distribution of Digg is more power-law-like, where only a handful of users have a dominant number of followers.

In both, we removed isolated nodes and those with less than 50 interactions in the log. Activation is defined as voting for the same article (Digg), or as watching or liking the same movie within 7 days (Flixster). These activation histories allow for learning of the activation probability without extra assumptions on the diffusion process, as opposed to the setting of some of the prior works, e.g., (Vaswani et al., 2017; Wu et al., 2019), where activations are synthetically simulated using a uniform distribution.

Constructing contextual feature vectors is not a trivial task. Our preliminary study showed that categorical features such as ZIP code and product categories lead to too much variance that washes away the similarity between users and between products. To avoid pathological issues due to such non-smoothness, we created user and product features using linear embeddings, as proposed in (Vaswani et al., 2017).

Specifically, for generating user features $\{\boldsymbol{x}_i \mid i \in \mathcal{V}\}$, we employed the Laplacian Eigenmap (Belkin & Niyogi, 2002) computed from the social graph $\mathcal{G}$, which is included in both Digg and Flixster datasets. Following (Vaswani et al., 2017), we used the eigenspectrum to decide on the dimensionality, which gave $d_1 = d_2 = 10$. For product (story or movie) features, we employed a probabilistic topic model (see, e.g. (Steyvers & Griffiths, 2007)) with the number of topics being 10. In this model, an product is viewed as a document in the "bag-of-votes" representation: Its $i$-th dimension represents whether the $i$-th user voted for the product. The intuition is that two articles should be similar if they are liked by a similar group of people. As a result, each product is represented by a 10-dimensional real-valued vector, which corresponds to a distribution over

Table 3: Methods compared. `IMFB` learns two node-wise coefficient vectors $\{(\boldsymbol{\beta}_i, \boldsymbol{\theta}_i)\}$. `IMLinUCB` learns only one coefficient vector $\boldsymbol{\theta}$. `DILinUCB` learns one node-wise coefficient vector $\{\boldsymbol{\beta}_i\}$. 'NA' denotes 'not available.'

| | score model $\bar{u}_{i,j}^{\boldsymbol{z}}$ | probability model $p_{i,j}^{\boldsymbol{z}}$ | source user | target user | product feature |
|---|---|---|---|---|---|
| `COIN` | $n_{i,j}^q$ | $n_{i,j}^q / N_{i,j}^q$ | NA | NA | $q = \text{category}(\boldsymbol{z})$ |
| `IMFB` | $\boldsymbol{\beta}_i^\top \boldsymbol{\theta}_j$ | $\text{proj}(\bar{u}_{i,j}^{\boldsymbol{z}} + \text{UCB})$ | NA | NA | NA |
| `IMLinUCB` | $\boldsymbol{x}_{i,j}^\top \boldsymbol{\theta}$ | $\text{proj}(\bar{u}_{i,j}^{\boldsymbol{z}} + \text{UCB})$ | $\boldsymbol{x}_{i,j} = \boldsymbol{x}_i \odot \boldsymbol{x}_j$ | | NA |
| `DILinUCB` | $\boldsymbol{x}_j^\top \boldsymbol{\theta}_i$ | $\text{proj}(\bar{u}_{i,j}^{\boldsymbol{z}} + \text{UCB})$ | NA | $\boldsymbol{x}_j$ | NA |
| `TensorUCB` | $\sum_{r=1}^{R} \prod_{l=1}^{D} \phi_l^\top \boldsymbol{w}^{l,r}$ | $\text{proj}(\bar{u}_{i,j}^{\boldsymbol{z}} + \text{UCB})$ | $\boldsymbol{\phi}_1 = \boldsymbol{x}_i$ | $\boldsymbol{\phi}_2 = \boldsymbol{x}_j$ | $\boldsymbol{\phi}_3 = \boldsymbol{z}$ |

the 10 latent topics identified. Once the product feature vectors are computed on a held-out dataset, they are treated as a constant feature vector for each product.

## 7.2 Baselines

Baseline methods were carefully chosen to comprehensively cover the major existing latent-modeling IM frameworks (see Introduction), as summarized in Table 3.

- `COIN` (Sarıtaç et al., 2016) learns the activation probability independently for each of the product categories $\{q\}$. The probability is computed based on the number of successful activations ($n_{i,j}^q$) and the total number of seed-selections ($N_{i,j}^q$). A control function is used for exploration-exploitation trade-off. To decide on $q$, we picked the most dominant topic (see Sec. 7.1), rather than the raw product labels for a fair comparison.

- `IMFB` (Wu et al., 2019) is a factorization-based method, where two node-wise coefficient vectors are learned from user response data without using user and product feature vectors.

- `IMLinUCB` (Wen et al., 2017) is an extension of the classical LinUCB to the edges as arms. This method requires an edge-level feature vector. The authors used the element-wise product of the user feature vectors as $\boldsymbol{x}_{i,j} = \boldsymbol{x}_i \odot \boldsymbol{x}_j$ in their experiments, which we used, too.

- `DILinUCB` (Vaswani et al., 2017) is another LinUCB-like algorithm that learns the coefficient vector in a node-wise fashion. The feature vector $\boldsymbol{x}_j$ in the table is the same as that of `TensorUCB`.

In addition, we implemented `Random`, which selects the seeds for a given round randomly.

Table 3 summarizes the high-level characteristics of the baseline methods. Since the number of seed nodes is almost always much smaller than $|\mathcal{V}|$, independently learning edge-wise or node-wise parameters is challenging in general. Such an approach requires so many explorations to get a reasonable estimate of the activation probability. `COIN`, `IMFB`, and `DILinUCB` are in this "overparameterized" category. On the other hand, `IMLinUCB` imposes a single regression coefficient vector $\boldsymbol{\theta}$ on all the edges. While this "underparameterized" strategy can be advantageous in capturing common characteristics between the edges, it cannot handle the heterogeneity over different products and over different types of user-user interactions. `TesnsorUCB` is designed to balance these over- and under-parameterized extremes: All the edges share the same coefficients $\{\boldsymbol{w}^{l,r}\}$, but they still have the flexibility to have $R$ different patterns. Importantly, `TesnsorUCB` is a nonlinear model that can capture user-user and user-product interactions through the product over $l$. See the first paragraph of Sec. 4 for a related discussion.

All the experiments used $K = 10$. `ORACLE` was implemented based on (Tang et al., 2014) with $\eta = 1 - \frac{1}{e} - 0.1$. For `TensorUCB`, we fixed $\sigma^2 = 0.1$ and the hyper-parameters $R$ and $c$ were optimized using an initial validation set of 50 rounds. As the performance metric, we reported the average cumulative expected regret $\frac{1}{t}R_t^\eta$ computed at each round. We reported on regret values averaged over five runs. For fair comparison, $\mathsf{P}_{\mathcal{G}}^*$ is computed by fitting a topic-aware IC model using maximum likelihood on the interaction logs as proposed in (Barbieri et al., 2013), whose parameterization is independent of any of the methods compared.

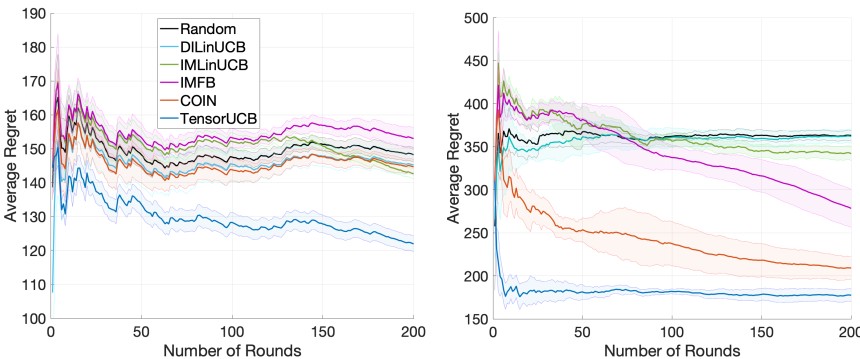

Figure 5: Average cumulative regret for Digg (left) and Flixster (right). Best viewed in colors.

## 7.3 Comparison of cumulative regret

As mentioned in Sec. 1, our original motivation for the tensor-based formulation is to capture the heterogeneity over different products. In our experiment, a new product (story or movie) is randomly picked (from 1 000 stories or 100 movies)at each campaign round $t$. In TensorUCB, $c$ was chosen from $\{10^{-3}, 10^{-2}, 10^{-1}, 1\}$, while $R$ was chosen in the range of $1 \leq R \leq 50$. We used the sigmoid function for proj$(\cdot)$ in Eq. (24).

Figure 5 compares TensorUCB against the baselines on Digg and Flixster. As is clearly shown, TensorUCB significantly outperforms the baselines. The difference is striking in Flixster, which has only 100 products (movies). Interestingly, TensorUCB captures the majority of the underlying preference patterns with only about 10 rounds of explorations. This is in sharp contrast to the "slow starter" behavior of COIN and IMFB, which can be explained by their "overparameterized" nature, as pointed out in Sec. 7.2. On the other hand, the Digg dataset includes as many as 1 000 products, and the interaction patterns in it may not have been fully explored in the relatively small number of rounds. This is consistent with the relatively small margin in the left figure. In addition, Digg's friendship network has a stronger power-law nature (Fig. 4). In such a network, seed users tend to be chosen from a handful of hub nodes, making room for optimization relatively smaller. Even in that case, however, TensorUCB captures underlying user preferences much more quickly than any other method.

In Flixster, COIN exhibits relatively similar behavior to TensorUCB. COIN partitions the feature space at the beginning, and hence, its performance depends on the quality of the partition. In our case, partition was done based on the topic model rather than the raw product label, which is a preferable choice for COIN. In Flixster, the number of products is comparable to the number of training rounds, which was 50 in our case. Thus the initial partitioning is likely to have captured a majority of patterns, while it is not the case in Digg. When the number of product types is as many as that of Digg, product-wise modeling can be unrealistic. In fact, many products (stories) ended up being in the same product category $q$ in Digg. Unlike the baseline methods, TensorUCB has a built-in mechanism to capture and generalize product heterogeneity directly through product context vector.

In the figure, it is interesting to see that the behavior of IMFB is quite different between the two datasets: It even underperforms Random in Digg while it eventually captures some of the underlying user preference patterns in Flixster. As shown in Table 3, IMFB needs to learn two unknown coefficient vectors at each node, starting from random initialization. Since it does not use contextual features, parameter estimation can be challenging when significant heterogeneity exists over users and products, as in Digg. Note that the previously reported empirical evaluation of IMFB (Wu et al., 2019) is based on simulated activation probabilities generated from its own score model and does not apply to our setting. It is encouraging that TensorUCB stably achieves better performance even under rather challenging set-ups as in Digg, indicating its usefulness in practice.

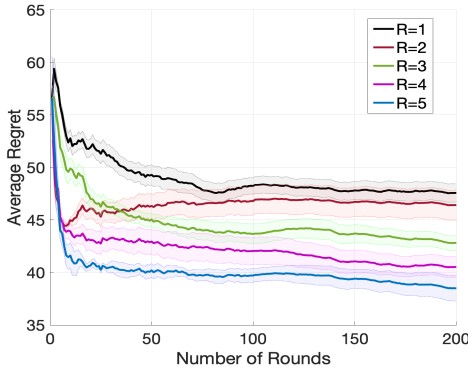
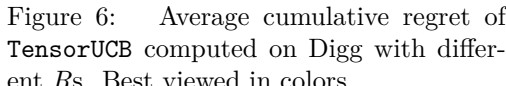

Figure 6: Average cumulative regret of `TensorUCB` computed on Digg with different $R$s. Best viewed in colors.

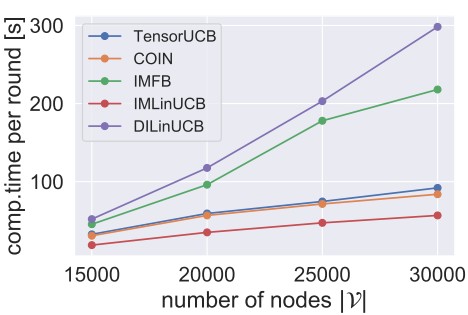

Figure 7: Computational time per round averaged over 200 rounds on Flixster. Best viewed in colors.

### 7.4 Dependency on tensor rank

`TensorUCB` uses the specific parameterization of CP expansion (6), whose complexity is controlled by the tensor rank $R$. Since, as discussed in Sec. 7.2, having $R > 1$ potentially plays an important role in handling heterogeneity in products and users, it is interesting to see how the result depends on $R$. Figure 6 shows how the average regret behaves over the first several values of $R$ on Digg. To facilitate the analysis, we randomly picked a single product (story) at the beginning and kept targeting the item throughout the rounds. In the figure, we specifically showed the first five $R$'s, where the change in the average regret was most conspicuous. In this regime, the average regret tends to improve as $R$ increases. Depending on the level of heterogeneity of the data, it eventually converges at a certain $R$ up to fluctuations due to randomness. The intuition behind Eq. (6) was that $R = 1$ amounts to assuming a single common pattern in the user preference while $R > 1$ captures *multiple* such patterns. This result empirically validates our modeling strategy.

### 7.5 Comparison of computational cost

Finally, Fig. 7 compares the computation time per round on Flixter, measured on a laptop PC (Intel i7 CPU with 32 GB memory). Error bars are negligibly small and omitted for clearer plots. To see the dependency on the graph size, we randomly sampled the nodes to create smaller graphs of $|\mathcal{V}| \approx 25\,000, 20\,000, 15\,000$. As shown in the figure, the computation time scales roughly linearly, which is understandable because the graph is quite sparse and the most nodes have a node degree that is much smaller than $|\mathcal{V}|$. `IMLinUCB` is fastest but significantly under-performed in terms of regrets in Fig. 5. `TensorUCB` is comparable to `COIN` and much faster than `IMFB` and `DILinUCB`. These results validate that `TensorUCB` significantly outperforms the baselines without introducing much computational overhead.

## 8 Concluding remarks

We have proposed `TensorUCB`, a tensor-based contextual bandit framework, which can be viewed as a new and natural extension of the classical UCB algorithm. The key feature is the capability of handling any number of contextual feature vectors. This is a major step forward in the problem of influence maximization for online advertising since it provides a practical way of simultaneously capturing the heterogeneity in users and products. With `TensorUCB`, marketing agencies can efficiently acquire common underlying knowledge from previous marketing campaigns that include different advertising strategies across different products. We empirically confirmed a significant improvement in influence maximization tasks, attributable to its capability of capturing and leveraging the user-product heterogeneity.

For future work, a more sophisticated analysis of the regret bound is desired. In particular, how to evaluate the variability due to the variational approximations of probabilistic tensor regression is still an open prob-

lem. From a practical perspective, how to define user and product feature vectors is a nontrivial problem. Exploring the relationship with modern embedding techniques would be an interesting future direction.

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

## Appendix

## A  Derivation of posterior distribution

This section explains how to solve Eq. (9). The the key idea of VB is to look at individual components $\{q^{l,r}\}$ of $Q$ one by one, keeping all the others fixed. For instance, for a particular pair of $(l,r) = (3,2)$, the minimization problem of Eq. (8) reads

$$
\min_{q^{3,2}} \left\{ \int d\boldsymbol{w}^{3,2}\, q^{3,2} \ln q^{3,2} - \int \prod_{l=1}^{D} \prod_{r=1}^{R} d\boldsymbol{w}^{l,r} q^{l,r} \ln Q_0(\boldsymbol{\mathcal{W}}) \right\},
$$

where $Q_0$ has been defined in Eq. (10). The main mathematical tool to solve this functional optimization problem is calculus of variations. A readable summary can be found in the appendix of Bishop (Bishop, 2006). Apart from deep mathematical details, its operational recipe is analogous to standard calculus. What we do is analogous to differentiating $x \ln x - ax$ to get $\ln x + 1 - a$, and equating it to zero. For a general $(l,r)$, the solution is given by:

$$
\ln q^{l,r} = \text{const.} + \langle \ln Q_0(\boldsymbol{\mathcal{W}}) \rangle_{\backslash (l,r)}, \tag{A.1}
$$

$$
\langle \ln Q_0(\boldsymbol{\mathcal{W}}) \rangle_{\backslash (l,r)} \triangleq \int \prod_{l' \neq l} \prod_{r' \neq r} d\boldsymbol{w}^{l',r'}\, q^{l',r'} \ln Q_0(\boldsymbol{\mathcal{W}}), \tag{A.2}
$$

where const. is a constant and $\langle \cdot \rangle_{\backslash (l,r)}$ denotes the expectation by $Q(\{\boldsymbol{w}^{l,r}\})$ over all the variables except for the $(l,r)$.

## A.1 Solution under the Gaussian model

Now let us derive an explicit form of the posterior. Using Eq. (10) for $Q_0$ together with Eq. (7) in Eq. (A.1), we have

$$\ln q^{l,r} = \text{const.} - \frac{1}{2}(\boldsymbol{w}^{l,r})^\top \boldsymbol{w}^{l,r} - \frac{1}{2\sigma^2}\sum_\tau \langle \{y_\tau - (\boldsymbol{\mathcal{W}}, \boldsymbol{\mathcal{X}}_\tau)\}^2 \rangle_{\backslash(l,r)}. \tag{A.3}$$

The expression (6) allows writing the last term in terms of $\{\boldsymbol{w}^{l,r}\}$:

$$\langle \{y_\tau - (\boldsymbol{\mathcal{W}}, \boldsymbol{\mathcal{X}}_\tau)\}^2 \rangle_{\backslash(l,r)} = \text{const.} - 2{\boldsymbol{w}^{l,r}}^\top \boldsymbol{\phi}_{\tau,l} \beta_\tau^{l,r} y_\tau^{lr}$$
$$+ {\boldsymbol{w}^{l,r}}^\top \boldsymbol{\phi}_{\tau,l} \boldsymbol{\phi}_{\tau,l}^\top \boldsymbol{w}^{l,r} \prod_{l' \neq l} \boldsymbol{\phi}_{\tau l'}^\top \langle {\boldsymbol{w}^{l'r}} {\boldsymbol{w}^{l'r}}^\top \rangle_{\backslash(l,r)} \boldsymbol{\phi}_{\tau l'}, \tag{A.4}$$

where $\beta_\tau^{l,r}$ and $y_\tau^{l,r}$ have been defined in Eq. (12).

Equations (A.3) and (A.4) imply that $\ln q^{l,r}$ is quadratic in $\boldsymbol{w}^{l,r}$ and thus $q^{l,r}$ is Gaussian. For instance, for $(l,r) = (3,2)$, collecting all the terms that depend on $\boldsymbol{w}^{3,2}$, we have

$$\ln q^{3,2} = \text{const.} + (\boldsymbol{w}^{3,2})^\top \frac{1}{\sigma^2}\sum_\tau (\beta_\tau^{3,2}\boldsymbol{\phi}_{\tau,3}) y_\tau^{3,r} - \frac{1}{2}(\boldsymbol{w}^{3,2})^\top \left\{ \mathsf{I}_d + \frac{1}{\sigma^2}\sum_\tau (\beta_\tau^{3,2}\boldsymbol{\phi}_{\tau,3})(\beta_\tau^{3,2}\boldsymbol{\phi}_{\tau,3})^\top \right\} \boldsymbol{w}^{3,2}.$$

By completing the square, we get the posterior covariance matrix $\boldsymbol{\Sigma}^{3,2}$ and the posterior mean $\bar{\boldsymbol{w}}^{3,2}$ as

$$\boldsymbol{\Sigma}^{3,2} = \left\{ \mathsf{I}_d + \frac{1}{\sigma^2}\sum_\tau (\beta_\tau^{3,2}\boldsymbol{\phi}_{\tau,3})(\beta_\tau^{3,2}\boldsymbol{\phi}_{\tau,3})^\top \right\}^{-1}, \qquad \bar{\boldsymbol{w}}^{3,2} = \frac{1}{\sigma^2}\boldsymbol{\Sigma}^{3,2}\sum_\tau (\beta_\tau^{3,2}\boldsymbol{\phi}_{\tau,3}) y_\tau^{3,r}, \tag{A.5}$$

which are the (batch-version of) solution given in the main text.

## B Derivation of the predictive distribution

This section explains how to perform the integral of Eq. (16) under

$$Q(\boldsymbol{\mathcal{W}}) = \prod_{l=1}^D \prod_{r=1}^R \mathcal{N}(\boldsymbol{w}^{l,r} \mid \bar{\boldsymbol{w}}^{l,r}, \boldsymbol{\Sigma}^{l,r}). \tag{B.6}$$

We first integrate w.r.t. $\boldsymbol{w}^{1,r}$. By factoring out $\boldsymbol{w}^{1,r}$ from the tensor inner product as $(\boldsymbol{\mathcal{W}}, \boldsymbol{\mathcal{X}}) = \sum_r (\boldsymbol{\phi}_1 b^{1,r})^\top \boldsymbol{w}^{1,r}$, we have

$$I_1 \triangleq \int \prod_{r=1}^R \mathrm{d}\boldsymbol{w}^{1,r} \, \mathcal{N}(\boldsymbol{w}^{1,r} \mid \bar{\boldsymbol{w}}^{1,r}, \boldsymbol{\Sigma}^{1,r})\mathcal{N}(u \mid (\boldsymbol{\mathcal{W}}, \boldsymbol{\mathcal{X}}), \sigma^2) = \mathcal{N}(u \mid u_1, \sigma_1^2),$$

where $b^{1,r} \triangleq (\boldsymbol{\phi}_2^\top \boldsymbol{w}^{2,r}) \cdots (\boldsymbol{\phi}_D^\top \boldsymbol{w}^{D,r})$ and

$$u_1 = \sum_{r=1}^R (\boldsymbol{\phi}_1^\top \bar{\boldsymbol{w}}^{1,r}) b^{1,r}, \quad \sigma_1^2 = \sigma^2 + \sum_{r=1}^R (b^{1,r}\boldsymbol{\phi}_1)^\top \boldsymbol{\Sigma}^{1,r}(b^{1,r}\boldsymbol{\phi}_1).$$

To perform the integral we used the well-known Gaussian marginalization formula. See, e.g., Eqs. (2.113)-(2.115) in Sec. 2.3.3 of Bishop (2006).

Next, we move on to the $l = 2$ terms, given $I_1$. Unfortunately, due to the nonlinear dependency on $\boldsymbol{w}^{2,r}$ in $\sigma_1^2$, the integration cannot be done analytically. To handle this, we introduce a mean-field approximation in the same spirit of that of the main text:

$$\sigma_1^2 \approx \sigma^2 + \sum_{r=1}^R (\beta^{1,r}\boldsymbol{\phi}_1)^\top \boldsymbol{\Sigma}^{1,r}(\beta^{1,r}\boldsymbol{\phi}_1), \tag{B.7}$$

where $\boldsymbol{w}^{2,r}, \ldots, \boldsymbol{w}^{D,r}$ have been replaced with their posterior means $\bar{\boldsymbol{w}}^{2,r}, \ldots, \bar{\boldsymbol{w}}^{D,r}$. The definition of $\beta_\tau^{l,r}$ is given by Eq. (12). We do this approximation for all $\sigma_1^2, \ldots, \sigma_D^2$ while keeping $u_1, \ldots, u_D$ exact.

As a result, after performing the integration up to $l = k$, we have a Gaussian $\mathcal{N}(u \mid u_k, \sigma_k^2)$, where

$$u_k = \sum_{r=1}^R \prod_{l=1}^k (\boldsymbol{\phi}_l^\top \bar{\boldsymbol{w}}^{l,r}) \prod_{l'=k+1}^D (\boldsymbol{\phi}_{l'}^\top \boldsymbol{w}^{l',r}), \qquad \sigma_k^2 = \sigma^2 + \sum_{l=1}^k \sum_{r=1}^R (\beta^{l,r} \boldsymbol{\phi}_l)^\top \boldsymbol{\Sigma}^{l,r} (\beta^{l,r} \boldsymbol{\phi}_l). \tag{B.8}$$

By continuing this procedure, we obtain the predictive mean and the variance in Eq. (17).

## C  Derivation of the regret bound

This section derives the upper bound given in Theorem 1 on the scaled regret defined in Eq. (25).

### C.1  Relationship with confidence bound

In Sec. 6, we have introduced the bounded smoothness condition (A1) and the monotonicity condition (A2). Let $\mathsf{P}_{t,\mathcal{G}} = [p_{t,i,j}^{\boldsymbol{z}}]$ be an estimate of the activation probabilities at the $t$-th round. Based on the monitonisity condition and the property of the `ORACLE`'s seed selection strategy, Wu et al. (2019) argued that the instantaneous regret (25) is upper-bounded as

$$\frac{1}{\eta} \mathbb{E}\left[ f(\mathcal{S}^*, \mathsf{P}_\mathcal{G}^*) - f(\mathcal{S}_t, \mathsf{P}_\mathcal{G}^*) \right] \leq \frac{1}{\eta} \mathbb{E}\left[ f(\mathcal{S}_t, \mathsf{P}_{t,\mathcal{G}}) - f(\mathcal{S}_t, \mathsf{P}_\mathcal{G}^*) \right] \tag{C.9}$$

with probability $1 - \delta$, where $\delta$ is the tail probability chosen in the UCB approach. Since $K \ll |\mathcal{V}|$ in general, it is possible for the event that may occur with possibility $\delta$ to play a significant role in the cumulative regret. Fortunately, Lemma 2 of (Wen et al., 2017), which lower-bounds the UCB constant $c$, guarantees that its contribution can be bounded by $\mathcal{O}(1)$, as argued by Wu et al. (2019). We will come back this point later to provide a condition on $c$ explicitly.

Combining with the smoothness condition (26) and the expression of UCB (24), we have

$$R_T^\eta \leq \frac{cB|\mathcal{V}|}{\eta} \sum_{t=1}^T \sum_{k=1}^K \sum_{l=1}^D \sum_{r=1}^R \kappa_{t(k),l,r} + \mathcal{O}(1), \tag{C.10}$$

$$\kappa_{t(k),l,r} \triangleq \sqrt{(\beta_{t(k)}^{l,r} \boldsymbol{\phi}_{t(k),l})^\top \boldsymbol{\Sigma}_{t(k)}^{l,r} (\beta_{t(k)}^{l,r} \boldsymbol{\phi}_{t(k),l})}, \tag{C.11}$$

where we have used the "$t(k)$" notation introduced in Sec. 3.2 in the main text. Here, $\boldsymbol{\Sigma}_{t(k)}^{l,r}$ is the covariance matrix computed using the data up to the $k$-th seed node in the $t$-th round. $\beta_{t(k)}^{l,r}$ and $\boldsymbol{\phi}_{t(k),l}$ are defined similarly. The summation over $j$ in Eq. (26) produces a constant of the order of node degree, which is bounded by $|\mathcal{V}|$.

Now our goal is to find a reasonable upper bound of $\kappa_{t(k),l,r}$. This term has appeared in the definition of the confidence bound $\mathtt{UCB}(\mathcal{X})$ in the main text. This is reminiscent of the regret analysis of vector contextual bandits (Dani et al., 2008; Chu et al., 2011; Abbasi-Yadkori et al., 2011), in which the analysis is reduced to bounding the vector counterpart of $\kappa_{t(k),l,r}$.

### C.2  Bounding confidence bound

Now that the cumulative scaled regret is associated with $\kappa_{t(k),l,r}$, let us prove the following Lemma, which supports the regret bound reported in the main text:

**Lemma 1.** *Under the assumption* $\|\beta_{t(k)}^{l,r}\phi_{t(k),l}\| \le 1,\ \forall l,r,t,k,$

$$\sum_{t,k,l,r} \kappa_{t(k),l,r} \le R\sqrt{\frac{TKD\sum_l d_l \ln\left(1 + \frac{TK}{d_l\sigma^2}\right)}{\ln\left(1 + \frac{1}{\sigma^2}\right)}}, \tag{C.12}$$

$$\le DR\sqrt{\frac{TKd\ln\left(1 + \frac{TK}{d\sigma^2}\right)}{\ln\left(1 + \frac{1}{\sigma^2}\right)}}, \tag{C.13}$$

*where* $d \triangleq \max_l d_l$.

*(Proof)* Under the $t(k)$-notation, the updating equation for the covariance matrix looks like

$$(\boldsymbol{\Sigma}_{t(k+1)}^{l,r})^{-1} = (\boldsymbol{\Sigma}_{t(k)}^{l,r})^{-1} + \left(\frac{\beta_{t(k)}^{l,r}}{\sigma}\right)^2 \phi_{t(k),l}\phi_{t(k),l}^{\top}, \tag{C.14}$$

which leads to an interesting expression of the determinant:

$$\det|(\boldsymbol{\Sigma}_T^{l,r})^{-1}| = \prod_{t=1}^{T-1}\prod_{k=1}^{K}\left(1 + \frac{\kappa_{t(k),l,r}^2}{\sigma^2}\right). \tag{C.15}$$

This follows from repeated applications of the matrix determinant lemma $\det|\mathsf{A}+\boldsymbol{a}\boldsymbol{b}^{\top}| = \det|\mathsf{A}|(1+\boldsymbol{b}^{\top}\mathsf{A}^{-1}\boldsymbol{a})$ that holds for any vectors $\boldsymbol{a},\boldsymbol{b}$ and invertible matrix $\mathsf{A}$ as long as the products are well-defined. Equation (C.15) implies $\det|\boldsymbol{\Sigma}_{t(k)}^{l,r}| \le 1$. By the assumption $\|\beta^{l,r}\phi_{t(k),l}\| \le 1$, we have $\kappa_{t(k),l,r} \le 1$.

Interestingly, the cumulative regret (C.10) can be represented in terms of the determinant. Using an inequality

$$b^2 \le \frac{1}{\ln(1 + \sigma^{-2})}\ln(1 + \frac{b^2}{\sigma^2}), \tag{C.16}$$

that holds for any $b^2 \le 1$, we have

$$\sum_{t=1}^{T}\sum_{k=1}^{K}\sum_{l=1}^{D}\sum_{r=1}^{R}\kappa_{t(k),l,r} \le \left[TKDR\sum_{t,k,r,l}\kappa_{t(k),l,r}^2\right]^{\frac{1}{2}},$$

$$\le \left[TKDR\sum_{t,k,r,l}\frac{\ln(1 + \frac{\kappa_{t(k),l,r}^2}{\sigma^2})}{\ln(1 + \sigma^{-2})}\right]^{\frac{1}{2}}, \tag{C.17}$$

$$= \left[TKDR\sum_{r,l}\frac{\ln\det|(\boldsymbol{\Sigma}_T^{l,r})^{-1}|}{\ln(1 + \sigma^{-2})}\right]^{\frac{1}{2}}, \tag{C.18}$$

where the last equality follows from Eq. (C.15).

The determinant is represented as the product of engenvalues. Since the geometrical mean is bounded by the arithmetic mean, we have

$$\det|(\boldsymbol{\Sigma}_T^{l,r})^{-1}|^{\frac{1}{d_l}} \le \frac{1}{d_l}\text{Tr}[(\boldsymbol{\Sigma}_T^{l,r})^{-1}], \tag{C.19}$$

$$= 1 + \frac{1}{d_l\sigma^2}\sum_{t=1}^{T-1}\sum_{k=1}^{K}\|\beta_{t(k)}^{l,r}\phi_{t(k),l}\|^2, \tag{C.20}$$

$$\le 1 + \frac{(T-1)K}{d_l\sigma^2}, \tag{C.21}$$

where the second equality is by Eq. (C.14) and the last inequality is by the assumption $\|\beta_{t(k)}^{l,r} \phi_{t(k),l}\| \leq 1$. Finally, we have

$$
\begin{aligned}
\sum_{t,k,r,l} \kappa_{t(k),l,r} &\leq \left[ \frac{TKDR^2}{\ln(1+\sigma^{-2})} \sum_{l=1}^{D} d_l \ln\left(1 + \frac{(T-1)K}{d_l \sigma^2}\right) \right]^{\frac{1}{2}}, \\
&\leq \left[ \frac{TKDR^2}{\ln(1+\sigma^{-2})} \sum_{l=1}^{D} d_l \ln\left(1 + \frac{TK}{d_l \sigma^2}\right) \right]^{\frac{1}{2}}.
\end{aligned} \tag{C.22}
$$

The final step for Eq. (C.13) is obvious. $\qquad\square$

As discussed above, to control the occurrence of tail events, the UCB constant $c$ (introduced in Eq. (24)) has to be lower-bounded. Using the expression (C.13) and following the same steps as in Lemma 3 in (Vaswani et al., 2017), we can show that, if

$$
c \geq DR\sqrt{\frac{Kd\ln\left(1 + \frac{TK}{d\sigma^2}\right)}{\ln\left(1 + \frac{1}{\sigma^2}\right)}} + \max_{l,r} \|\boldsymbol{w}^{l,r}\|_2, \tag{C.23}
$$

the regret upper bound (28) holds with probability at least $1 - \delta$.[1]

---

[1]We thank an anonymous reviewer for pointing out the need for explicitly stating the condition on $c$.

