# OpenReview forum: "Targeted Advertising on Social Networks Using Online Variational Tensor Regression"
_TMLR — Rejected by TMLR_

### Review · Reviewer_NgRk · 2022-06-11

**Summary Of Contributions:**

This paper uses tensor-based contextual bandits to handle a online influence maximization problem. An online variational algorithm with a mean-field approximation is proposed.

**Requested Changes:**

Major comments are in previous section.

Some minors:
1. In the beginning of Section 4 when giving an example of D=3 for H_w, there is no definition of H_w so far. And it's hard to understand how linUCB fails to model their interactions and what is the consequence.
2. In (3), why W is a random variable.
3. Do you restrict on Gaussian noise?
4. For the UCB policy in (24), what is J? And why the LHS depends on (i, j) but RHS does not depend on (i, j)?
5. What is ORACLE? In the algorithm section, there is no definition.

**Strengths And Weaknesses:**

The theorem is very rough and not rigorous. What do you mean "UCB(X ) in Eq. (24) is exact"? I suggest the author refer to other published bandit literatures to revise their theorem. Please clearly list your assumptions. For me, the regret bound is questionable. How do you handle the estimation error from your tensor regression methods? Do you consider Bayesian regret or frequentist regret since you assume W is a random variable.

There is also no careful comparison on regret bounds with existing literature. How do you compare your regret bounds with Hao et al., 2020 and  Jun et al., 2019a on their problems since this paper claims to consider a more general case? How is the comparison with naively implementing Jun et al., 2019a's algorithm (unfolding a tensor into a matrix)? How does it compare with existing IM literatures? Do you assume the same assumptions? I think this is required in the bandit literature and could help the reader understand how the low-rank structure is used. From the current bound, I can not see how the tensor low-rank structure helps.

I think the tensor regression model in Section 4.1 about influence maximization should be carefully interpreted. Why does the observation also has the low-rank structure? I mean why you can assume the context feature in (4) can have such rank-1 structure. This is not typical even in standard supervised tensor regression where usually the coefficient is assumed to have low-rank. And also the low-rank coefficient needs to be interpreted well under the current problem since it greatly restricts the parameter space which could add unrealistic constraints (for example, is it reasonable to assume W is rank-1?).

Section 4.2 and 4.3 are purely supervised learning. Is this completely new for tensor regression? If not, it is better to acknowledge this since I have seen multiple Bayesian tensor regression literature.

---

> ### Author Response · Authors · 2022-06-12
> **Claimed goal of the paper**
>
> Thank you very much for your detailed comments.  Before getting into the technical detail, we would like to remind the reviewers of three points:
> 1. The claimed goal of this paper is to handle the heterogeneity of marketing campaigns in products and users.  The probabilistic tensor regression is a new solution approach.
> 1. Unlike vector-based linear regression, probabilistic tensor regression has no known exact analytic solution.
> 1. Because of 2, deriving a mathematically exact regret bound is still an open problem; we are not claiming an accomplishment in the regret analysis.
>
> We totally understand the reviewer's wish for a regret analysis that covers the possible variability due to the approximations we used, i.e.,
> - Variational Bayes (VB) approximation, and
> - Mean-field (MF) approximation.
>
> However, exact bounding these approximations is challenging and still an open problem. We will definitely improve the description to make this point much clearer.
>
> (Detailed comments follow below)

---

> ### Author Response · Authors · 2022-06-12
> **[Assumptions of regret analysis]**
>
> Related to [2 and 3 above](https://openreview.net/forum?id=wduAKrhm5w&noteId=RTm0tekXXG),
> > "assuming UCB(x) is Eq.(24) is exact"
>
> means that
> >"assuming that the derived posterior distribution for $\mathcal{W}$ is an exact analytic solution without the VB and MF approximations."
>
> We wish we could estimate the impact of the approximations precisely, but it is still an open problem.  Hence, we are afraid that "careful comparison" with Hao et al., etc., is not within the scope of the paper.  The cited works use a simpler setting that does not need such approximations.  Hence, they are not directly comparable in terms of regret bound.
>
> We will definitely update the paper to clarify this point.

---

> ### Author Response · Authors · 2022-06-12
> **[Direct product form and CP expansion]**
>
> Your point must be:  "*We never use any approximation in probabilistic linear regression for vectors.  Why do we have to use such unnatural assumptions?*"
>
> We understand your concern.  The [points 2 and 3 above](https://openreview.net/forum?id=wduAKrhm5w&noteId=RTm0tekXXG) are again important.  Bayesian tensor regression has **no known exact analytic solution**; there does not exist a convenient probability distribution for tensors, either.  If you didn't make any assumptions, you would not be able to derive an explicit posterior distribution for $\mathcal{W}$. This is exactly why we had to introduce the direct product from (4) and CP expansion in Eqs. (5)-(6). Although we have looked at other formulations, the model presented in the paper was the only practically feasible approach.
>
> In the first paragraph of Section 4, we have briefly discussed why we used that particular direct-product form.  It can be viewed as the simplest nontrivial model beyond the simple concatenation approach.  As seen from Eqs. (11)-(12), the direct product form (4) yields nonlinear terms for the posterior mean.  This is how our model captures the affinity between user pairs and products.
>
> We will definitely update Section 4 to clarify this point.

---

> ### Author Response · Authors · 2022-06-12
> **[Novelty of the proposed Bayesian tensor regression approach]**
>
> We have discussed this point in the third paragraph of Section 2.  As highlighted *in italic*, there are two main differences in our problem setting from the previous Bayesian tensor regression works.
> - We need to get an explicit form of the posterior distribution for evaluating the UCB.
> - We need to get an online updating learning algorithm.
>
> To the best of our knowledge, the method presented in the paper is the first that has these two features.  We will definitely update Section 3 to clarify this point.

---

> ### Author Response · Authors · 2022-06-12
> **[Answer to the minor comments]**
>
> Here are our reply to your comments. We thank you again for your detailed comments!
>
> 1. We are sorry for the unclear description. The vector concatenation approach in the first paragraph in Section 4 refers to LinUCB; hence, we assumed the standard vector-based linear regression model for $H_\mathcal{W}$.  We will make this point more explicit in version 2.
>
> 2. We understand your point.  In our Bayesian tensor regression, W is treated as a random variable, as explicitly shown in Eq. (7).
>
> 3. We follow the (Bayesian counterpart of) LinUCB regarding the noise model, which is Gaussian.
>
> 4. $J(\cdot)$ is defined in the sentence just below Eq. (4).  It is a mapping function to map the output of the linear regression onto  [0,1].  We totally agree that the dependency on $(i,j)$ in Eq. (24) is confusing.  We are sorry.  We will make this explicit in version 2:  In Section 4.2, we have mentioned *"$\phi_{\tau,1}$ and $\phi_{\tau,2}$ are used for the node feature vectors, serving as the proxy for the node indexes''.* In (24), the first two of $D$ feature vectors carry the information of the $i$- and $j$-th users.
>
> 5. We have described ORACLE in the first part of Section 3.  We will add a reference to this part in later sections for readers' convenience.

---

> ### Author Response · Authors · 2022-06-29
> **Summary of revisions in ver.2**
>
> Thank you again for your detailed comments. We have updated the paper according to your comments. Here is a summary.
>
> **[Claimed goal]**
> - We have clearly mentioned that deriving an exact bound for variational approximations is an open problem before Theorem 1 in Section 6 (Regret analysis).
>
> - We have also added a concluding remark about the open problem.
>
>
> **[Novelty of the proposed Bayesian tensor regression]**
>
> - We have commented on the need for an analytic form of predictive distribution for UCB in the paragraph on tensor regression in Section 2 (related work).
>
> - We have added an extra comment in Section 4.1 to highlight (1) that Hao et al.(2020) is not contextual and is not comparable to our model and (2) that Jun et al. (2019) is a bilinear model lacking the capability of handling more than two feature vectors:
>
> > Note that Eq.(4) includes the regression models in *bilinear* bandits (e.g. (Jun et al., 2019a)) and *non*-contextual tensor bandits (Hao et al., 2020) as special cases. Specifically, bilinear models can handle only the $D=2$ case, while our model can handle $D \geq 3$. *Non*-contextual tensor regression cannot accommodate the feature vectors {$\{ \phi_l \in \mathbb{R}^{d_l} \mid l=1,\ldots,D \}$}.
>
>
> **[Assumptions of regret analysis]**
>
> - We have listed the assumptions used to derive the regret bound in Section 6. The assumption (A3) is about the variability of the variational approximations.
>
> - We have provided a concluding remark that bounding the variability of the variational approximations is an interesting future research topic.
>
>
> **[Direct product form and CP expansion]**
>
> - We have commented on the implication of the particular choice of the CP expansion right after Eq.(6). The point is that CP expansion seems to be the only approach allowing derivation of an analytic form of predictive distribution, to the best of our knowledge.
>
> - We have also added a comment on the practical importance of having $R>1$ in capturing heterogeneity in products in Sections 4.1, 7.2, and 7.4.
>
>
> **[Other points]**
>
> - Regarding the limitation of the vector concatenation approach, we have mentioned that a linear regression model is assumed for $H_{\mathcal{W}}$ in the LinUCB-like model in the first paragraph of Section 4.
>
> - We have commented that the noise model in Eq.(3) is Gaussian in Section 3.2.
>
> - We have changed notation $J(\cdot)$ to the commonly used proj$(\cdot)$ to avoid confusion.
>
> - We have changed the notation $p_{i,j}$ to $p_{i,j}^{\mathbf{z}}$ to show the dependency on the product feature $\mathbf{z}$. We have also updated Table 1 to highlight that $\phi_1$ and $\phi_2$ are allocated to the source and target user features while $\phi_3$ is allocated to the product feature.
>
> - We have added a new subsection ``7.2 Baselines'' to highlight the implications of the low-rank approximation of tensors. As described there as well as in Sections 7.3 and 7.4, having $R>1$ amounts to allowing multiple patterns of products.
>
> - We have added a reference to Oracle in Section 5.3 (Algorithm summary).

---

### Review · Reviewer_JZxF · 2022-06-21

**Summary Of Contributions:**

The paper deals with the problem of online influence maximization in social networks. In particular, it introduces a methodology that allows to capture the heterogeneity among users and products through a tensor regression framework. Contrary to previous contextual bandit approaches that are restricted to user node features, the tensor model allows capturing interactions among users and products. The paper further introduces a variational learning approach of the tensor regression model. The proposed TensorUCB algorithm is presented in a rigorous manner. It is also important to see that a regret analysis is presented. The proposed model is evaluated on two real-world datasets, and its performance is compared against various baseline models for online influence maximization.

**Broader Impact Concerns:**

No broader impact concerns were identified.

**Requested Changes:**

+ To motivate the proposed work, the authors mention *“The lack of capability to simultaneously deal with heterogeneity over products and user pairs capturing their interactions.”*. At this part of the paper, this limitation statement is not clear. I will propose the authors give an example, making this argument more concrete.

+ In the empirical analysis, two graph datasets have been used. However, there is no discussion about the potential impact of the graph structural properties on the performance of the proposed model. Further understanding of the behavior of the model with respect to the degree heterogeneity that the graph datasets show, would be helpful at this point. I would propose to the authors to investigate this aspect.

+ In the experiments regarding the average regret shown in Fig. 4, it turns out that IMFB is among the worst-performing models. Is there a plausible explanation for that? Looking at the corresponding experiments of the IMFB KDD ’19 paper (specifically, Fig. 2), IMFB’s reward suggest a totally different picture – compared to IMLiUCB, DILiUCB. The datasets though are different, which could further support my previous comment regarding the impact of the structural properties of the graph.

+ Since IMFB/LinUCB do not leverage product features, how was exactly the setup? Did the authors try to concatenate the user and product vectors?


**Strengths And Weaknesses:**

**Strengths**
+ The paper studies an interesting formulation of the online influence maximization problem.

+ Solid and clear presentation of the proposed models.

**Weaknesses**
 + In my view, the main weak point of the manuscript has to do with the empirical analysis (see further comments below).

---

> ### Author Response · Authors · 2022-06-29
> **Response to 1: Clearer statement of motivation**
>
> We totally agree that the original submission should have been clearer about limitations of the existing work. We have made that part in Section 1 simpler and clearer:
>
> > Although encouraging results have been reported in these works, there is one important limitation that restricts their usefulness in practice: *Absence of capability to incorporate product features.* This is critical in practice since marketing campaigns typically include many different products and strategies applied to a diverse population, and different types of products are expected to follow different information diffusion dynamics.

---

> ### Author Response · Authors · 2022-06-29
> **Response to 2: Analyzing degree heterogeneity of the datasets**
>
> We agree that explicit reference to the degree distribution helps the reader understand the general properties of the datasets. We have newly added Figure 4 for out-degree distributions and Table 2 to summarize statistics of the social networks.
>
> The figure suggests that Digg's network follows a power-law distribution more strongly than Flixster, and hence, there are only a handful of dominant influencers in the network. This explains why improvement by TensorUCB is relatively small in Digg as compared to Flixster.

---

> ### Author Response · Authors · 2022-06-29
> **Response to 3: Explaining IMFB's performance more clearly**
>
> We understand your point. The biggest reason why (Wu et al., 2019) could get an excellent result in their Fig.2 is that they had synthetically generated the activation probabilities using their own model. Their Section 5.1 says,
>
> > "The activation probabilities on the edges are generated according to Eq (1)." (Section 5.1 of (Wu et al., 2019))
>
> Here Eq.(1) is their score model that is recapitulated in our Table 3 in the revised version. In our setting, the datasets have activation logs, and the activation probabilities were learned from scratch.
>
> We have created a new subsection 7.2 to explain the baseline methods in comparison to the proposed method TensorUCB. The limited performance of IMFB can be explained by
>
> 1. its (over) parameterization and
> 2. the lack of capability of accommodating contextual information.
>
> We have updated Section 7.3 to make this point clearer.

---

> ### Author Response · Authors · 2022-06-29
> **Response to 4: Clearer description of experimental setup**
>
> We agree that the original version did not provide a clear-cut explanation of the baselines. We have significantly updated Section 7. Specifically, we have added a new subsection 7.2, where Table 3 summarizes the baseline methods with a clear reference to the contextual information used in each of the baseline methods.
>
> Regarding the vector concatenation strategy for incorporating product features, we have provided a brief discussion at the beginning of Section 4. In the LinUCB-like setting, where the activation probability is evaluated via a linear regression model, concatenating product and user features may not be a promising approach because the affinity between users and the product is not effectively represented.

---

### Review · Reviewer_GMUM · 2022-06-21

**Summary Of Contributions:**

This paper studied online influence maximization (IM) for targeted advertising, with the goal of selecting best $k$ seed nodes (users) to propagate advertisement via social network. Online IM is commonly formulated as contextual bandits and the authors proposed a new tensor-based contextual bandit formulation. Previous works mostly considered context information of users and assumed the activation probability between two users is a linear model determined by the pair of user features and unknown model parameters. The motivation of this paper is to formulate richer context information by a tensor, allowing more than 2 context features to be leveraged, e.g., an additional feature representing the product or marketing strategy.

Following this formulation, the authors proposed a tensor regression based contextual bandit solution for IM. The authors introduced a tractable online updating method based
on a variational mean-field approximation of the tensor regression. The approximation allows the authors to analyze the Upper Confidence Bound (UCB) of the activation probability estimate, which leads to the proposed TensorUCB algorithm. Sublinear regret is provided for the proposed method. Empirical evaluation on two real-world graphs showed that TensorUCB outperforms the baselines methods.

**Requested Changes:**

  1. [Critical] Please clarify the problem formulation and assumptions as mentioned in Weaknesses 1.
  2. [Critical] Please fix the error in regret analysis.
  3. [Critical] Please justify the smoothness assumption.
  4. The experiment description should be clarified and comparison could be made in a fair way.
  5. The authors are suggested to compare regret in Theorem 1 with others baselines' regret bounds in detail.

**Strengths And Weaknesses:**

Strengths:

  1. The main contribution is the new tensor formulation that includes additional context features such as product features, which is a more realistic formulation.

  2. TensorUCB applied mean-field approximation for online variational tensor regression, which is another contribution and of independent interest. While the novelty/contribution of confidence bound analysis is limited because of ignoring the approximation error, the idea of TensorUCB algorithm could still be interesting to the readers.

  3. Empirical results suggested that incorporating additional contextual information helps the TensorUCB algorithm learns activation probability faster and reduces regret.

Weaknesses:

  1. The problem formulation and assumptions are not clearly explained and need significant improvement. This include:
    1.1 The first confusion is whether the activation probabilities $\{p_{i,j}^*\}$ are fixed or changing. The notation suggested fixed probabilities, but the experiment and algorithm description suggested that the probabilities are different at different rounds depending on product context. While in Algorithm 1 the product context $\phi_3$ is fixed over time, in experiment the features are different for different products: ``the product is randomly picked at each campaign round $t$''. Then the activation probabilities could be different at different rounds following the tensor model, where the user features and the susceptibility tensor $\mathcal{W}$ are fixed. The authors should make the notation, algorithm description and experiment setting consistent.
    1.2 The author should clarify that this paper assumed edge-level feedback as opposed to others studies on node-level feedback.
    1.3 The diffusion model needs to be clarified. This paper follows the independent cascade model as opposed to linear threshold model or model-independent assumption.
    1.4 Noise in Eq(5) is not specified. From the appendix the noise is assumed to be gaussian, which should be specified earlier.
    1.5 All assumptions should be clearly stated in the main paper, e.g., Problem Setting section.  For example, monotonicity assumption in Appendix C.1 should be mentioned earlier.

  2. Error in regret analysis. Note that the high probability UCB analysis in Eq(24) only holds for a single data point. When analyzing the regret and summing over $T$ rounds, the probability of all predictions within their UCB requires a union bound. So if the regret has $1-\delta$ probability holds for the first term in Eq(C.9), then the UCB of each predictive distribution should be $1- \delta/(TK)$, which means the coefficient $c$ in Eq(24) should not be a constant but related to $T, K$, which will change the order of the regret in Theorem 1. Similar union bound is used in Chu et al., 2011 and the authors are suggested to fix the proof error accordingly. This problem is different from the linear bandit analysis where the confidence ellipsoid holds for all $t$ with probability $1-\delta$ (see Abbasi-Yadkori et al., 2011).

  3. Smoothness assumption in  Eq(26) is not justified. A commonly used smoothness assumption for IM ( Chen et al., 2016, Wu et al., 2019) is $\sum (i,j) \in \mathcal{E}$ instead of only summing over $i \in S_t$ in this paper, because the smoothness should be related to the whole graph instead of only edges connected to the seed nodes. A potential extra $|V|$ dependency might be introduced if the smoothness assumption is changed to the commonly used one. The authors should provide justification of this assumption or revise the assumption and the regret analysis accordingly.

  4. Experiment description is not clear. The paper did not clarify the inputs to different algorithms, e.g., what context features for IMLinUCB and DILinUCB. Also the comparison may not be fair if the product features are changing since previous works assumed static activation probabilities. A (potentially fair) comparison is to build a new bandit instance for each product's campaign.

---

> ### Author Response · Authors · 2022-06-29
> **Response to 1: Improving overall clarity**
>
> We thank the reviewer for detailed comments. We have addressed all the issues you raised in the revised version as follows.
>
> **[1.1: Clearer notation on the dependency of activation probability]**
>
> We are sorry that the notation was quite confusing. As you suspected, $p_{i,j}$ depends not only on the user features $\mathbf{x}_i,\mathbf{x}_j$ but also on the product feature $\mathbf{z}$ (and possibly other feature vectors $\mathbf{\phi}_4,\ldots,\mathbf{\phi}_D$). To fix this issue, we have:
>
> - Changed the notation from $p_{i,j}$ to $p_{i,j}^{\mathbf{z}}$ although we started with $p_{i,j}$ in Section 1 for consistency with the literature.
> - Updated Table 1 to clarify that $\mathbf{\phi}_1,\mathbf{\phi}_2$ are allocated to the user feature vectors, and $\mathbf{\phi}_3$ is allocated to the product feature vector.
> - Added a brief description of the notation in Sections 3.1, 3.2, and 5.2 for readers' convenience.
> - Added Table 3 to explain our notation in light of the baseline methods.
>
>
>
> **[1.2: Clearer reference to the use of edge-level feedback ]**
>
> In addition to the improvement of the notation as above, we have commented that the model utilizes edge-level feedback in Section 1. We have also added Table 3 to help the reader understand the difference and commonality between the baseline methods and the proposed method.
>
>
> **[1.3: Clearer reference to diffusion model]**
>
> We have added an explicit reference to the independent cascade model  in the beginning of Section 3. The activation probability in our work is defined as the probability that a source user influences a target user into buying (or voting, liking, etc.) a product. It can be viewed as a proxy of the underlying diffusion dynamics. We have updated our comment on this point there for better clarity.
>
> We have also added a brief description in Section 7.1 on the empirical evaluation setting, which does not rely on a specific diffusion model (apart from the black-box ORACLE), because interaction logs in our datasets allow estimating the ground truth activation probabilities.
>
>
> **[1.4: Clearer reference to noise model]**
>
> We have added a sentence saying that the noise will be Gaussian in the paragraph just below Eq.(3).
>
>
> **[1.5: Clearer reference to the assumptions of Theorem 1]**
>
> Before the theorem, we itemized the four assumptions (A1)-(A4) we made.

---

> ### Author Response · Authors · 2022-06-29
> **Response to 2: Corrections to the regret bound proof**
>
> We appreciate the reviewer for bringing up this issue. We agree that the constant $c$ has to be lower-bounded, and the bound should be related to $T$ (the total number of rounds) and $K$ (seed budget). We have added the lower-boundedness condition of $c$ in Theorem 1 and its proof.
>
> In the original submission, we referred to Lemma 2 of (Wen et al., 2017), which states that $c$ is lower-bounded by a quantity related to $T$ in our notation. That condition was required to control the occurrence of tail events. However, we did not explicitly mention the bound of $c$. We totally agree that the dependency of $c$ on $T$ and $K$ should be made explicit.

---

> > ### Comment · Reviewer_GMUM · 2022-07-26
> > **New regret upper bound is linear**
> >
> > Thank you for the revision! I noticed that in the revised Theorem 1 Eq(27), $c$ is lower bounded by $\sqrt{TKd}$ and the regret upper bound will be *linear*. I am not sure whether that is a typo or not -- Lemma 3 in (Vaswani et al., 2015) shows a $\sqrt{d|V|}$ lower bounded instead. The derivation of $c$ is not clear, but this does not seem to be a typo to me.

---

> > > ### Author Response · Authors · 2022-07-26
> > > **It was a typo**
> > >
> > > Thank you for pointing it out. This is indeed a critical point. It was, in fact, a typo! The derivation follows the same steps as (Vaswani et al., 2017), and the bound is not proportional to $\sqrt{T}$. We will correct the typo and post a revised version soon later.

---

> ### Author Response · Authors · 2022-06-29
> **Response to 3: Double summation in the smoothness condition**
>
> We are sorry that our unclear notation seems to have been confusing. The summation was in fact a double summation. To clarify this point, we have rewritten the condition as
> $$
> \cdots \leq B \sum_{i \in \mathcal{S_t}} \sum_{j \sim i}  |\cdots |
> $$
> in the revised paper. This summation was written confusingly as
> $$
> \cdots \leq B \sum_{i \in \mathcal{S}_t, j \sim i}  |\cdots |
> $$
> in the original submission.

---

> ### Author Response · Authors · 2022-06-29
> **Response to 4:  Clearer description of baselines**
>
> We have significantly updated Section 7 by creating a new subsection ("7.2 Baselines"). We have added Table 3 to clearly explain the difference and commonality of the methods compared, especially in terms of the contextual information they use.
>
> As you point out, one straightforward approach to handling product heterogeneity is to create different models for each product (i.e., "new bandit instance"). This is what COIN does. In the current setting, however, the number of products is as large as 1000; the Naïve use of raw product labels for product-wise modeling can be unrealistic. For COIN, we provided product partitioning based on a topic model rather than raw product labels. This is a preferable choice for COIN. Even in that case, we have confirmed that COIN underperforms the proposed method.
>
> We have also discussed at the beginning of Section 4 why the simple vector concatenation approach would not work for incorporating product features in the LinUCB-like setting.

---

### Author Response · Authors · 2022-07-06
**Happy to answer further questions**

Dear Reviewers,

We believe the revised version 2 addresses all the questions/comments that have been raised. Of course we'd be happy to answer further questions, if there are any.

We’d also like to take this opportunity to remind the reviewers of the **paper's main technical claims**. The motivating problem we tackle in this paper is how to incorporate both user and product features in the context of online advertising. Specifically,

1. We have introduced a tensor-based contextual bandit model, which we believe is a new solution approach.
2. Since tensor regression does not have a closed-form solution, we have developed an online variational algorithm that permits the derivation of an analytic form of the predictive distribution, thereby leading to a tensor version of the UCB algorithm, which enjoys practical benefits.

We believe that these claims are supported by the corresponding analytic derivation and empirical evaluation.

For comparison purposes, we have also derived a regret bound for TensorUCB. Since our problem setting is novel, it is based on certain new assumptions related to approximation errors in tensor regression. The revised version mentions that lifting those assumptions would be an interesting research direction to pursue in the domain.


Sincerely,

The authors

---

### Decision · Action_Editors · 2022-08-05

**Recommendation:** Reject

**Comment:**

The paper proposed an interesting tensor-based contextual bandit framework for online targeted advertising. The reviewers are grateful to the authors that they have made efforts in improving the presentation and the experimental part in the revised version. However, during the lengthy discussion, most of the reviewers still have the main concern about the rigorous evidence in its regret analysis. In particular,  the current theorem (Theorem 1) requires the assumption of "UCB(X ) in Eq. (24) is exact" which can be too strong and might not hold true.  As two out of three reviewers are leaning toward rejection, I recommend its rejection.